# An Intrinsic Host Defense against HSV-1 Relies on the Activation of Xenophagy with the Active Clearance of Autophagic Receptors

**DOI:** 10.3390/cells13151256

**Published:** 2024-07-26

**Authors:** Camila Pino-Belmar, Rayén Aguilar, Guillermo E. Valenzuela-Nieto, Viviana A. Cavieres, Cristóbal Cerda-Troncoso, Valentina C. Navarrete, Paula Salazar, Patricia V. Burgos, Carola Otth, Hianara A. Bustamante

**Affiliations:** 1Instituto de Microbiología Clínica, Facultad de Medicina, Universidad Austral de Chile, Valdivia 5110566, Chile; cpinobelmar@gmail.com (C.P.-B.); rayenaguilar95@gmail.com (R.A.); valentina.navarrete@alumnos.uach.cl (V.C.N.); paula.salazar@uach.cl (P.S.); 2Instituto de Medicina, Facultad de Medicina, Universidad Austral de Chile, Valdivia 5110566, Chile; guillermo.valenzuela@uach.cl; 3Centro Interdisciplinario de Estudios del Sistema Nervioso (CISNe), Universidad Austral de Chile, Valdivia 5110566, Chile; 4Organelle Phagy Lab, Centro de Biología Celular y Biomedicina (CEBICEM), Facultad de Medicina y Ciencia, Universidad San Sebastián, Lota 2465, Santiago 7510157, Chile; viviana.cavieres@uss.cl (V.A.C.); cristobal.cerda@kuleuven.be (C.C.-T.); patricia.burgos@uss.cl (P.V.B.); 5Departamento de Ciencias Biológicas y Químicas, Facultad de Medicina y Ciencia, Universidad San Sebastián, Lota 2465, Santiago 7510157, Chile; 6Centro Científico y Tecnológico de Excelencia Ciencia & Vida, Santiago 7750000, Chile

**Keywords:** HSV-1, autophagy, autophagic receptor, intrinsic host defense, xenophagy, US11

## Abstract

Autophagy engulfs cellular components in double-membrane-bound autophagosomes for clearance and recycling after fusion with lysosomes. Thus, autophagy is a key process for maintaining proteostasis and a powerful cell-intrinsic host defense mechanism, protecting cells against pathogens by targeting them through a specific form of selective autophagy known as xenophagy. In this context, ubiquitination acts as a signal of recognition of the cargoes for autophagic receptors, which direct them towards autophagosomes for subsequent breakdown. Nevertheless, autophagy can carry out a dual role since numerous viruses including members of the *Orthoherpesviridae* family can either inhibit or exploit autophagy for its own benefit and to replicate within host cells. There is growing evidence that Herpes simplex virus type 1 (HSV-1), a highly prevalent human pathogen that infects epidermal keratinocytes and sensitive neurons, is capable of negatively modulating autophagy. Since the effects of HSV-1 infection on autophagic receptors have been poorly explored, this study aims to understand the consequences of HSV-1 productive infection on the levels of the major autophagic receptors involved in xenophagy, key proteins in the recruitment of intracellular pathogens into autophagosomes. We found that productive HSV-1 infection in human neuroglioma cells and keratinocytes causes a reduction in the total levels of Ub conjugates and decreases protein levels of autophagic receptors, including SQSTM1/p62, OPTN1, NBR1, and NDP52, a phenotype that is also accompanied by reduced levels of LC3-I and LC3-II, which interact directly with autophagic receptors. Mechanistically, we show these phenotypes are the result of xenophagy activation in the early stages of productive HSV-1 infection to limit virus replication, thereby reducing progeny HSV-1 yield. Additionally, we found that the removal of the tegument HSV-1 protein US11, a recognized viral factor that counteracts autophagy in host cells, enhances the clearance of autophagic receptors, with a significant reduction in the progeny HSV-1 yield. Moreover, the removal of US11 increases the ubiquitination of SQSTM1/p62, indicating that US11 slows down the autophagy turnover of autophagy receptors. Overall, our findings suggest that xenophagy is a potent host defense against HSV-1 replication and reveals the role of the autophagic receptors in the delivery of HSV-1 to clearance via xenophagy.

## 1. Introduction

Herpes simplex virus type 1 (HSV-1) is the vernacular name for the species formally named *Simplexvirus humanalpha1*. This virus belongs to the family *Orthoherpesviridae*, subfamily *Alphaherpesvirinae*, genus *Simplexvirus*, according to the International Committee on the Taxonomy of Viruses (ICTV) [1], and is one of eight known Herpesviruses that naturally only infect humans (HHVs) [2]. HSV-1 is the most common HHV and is highly widespread [3] with a worldwide prevalence of 67% in the population under 50 years old, has no seasonal variation, and infects both sexes equally [3,4,5]. HSV-1 is frequently acquired during childhood [3] where the most common clinical manifestations are fluid-filled blisters that appear suddenly on oral and pharyngeal mucosa membranes (Gingivostomatitis) or in lips, commonly known as “cold sores” (Herpes Labialis), appearing on the vermillion border of the lips [2,3]. However, the infection may result in more serious complications, including birth defects in neonates (Neonatal Herpes), ocular disease, blindness (Herpetic Keratoconjunctivitis), and brain and meninges inflammation (Herpes Meningoencephalitis). Moreover, elderly people, immunocompromised people, or those with comorbidities can be strongly affected by an increase in the frequency of HSV-1 reactivation, as this is one of the most important causes of death by encephalitis in adults globally [2,6]. These clinical manifestations are a consequence of its productive replication in keratinocytes of skin and mucous membranes [2,3], followed by the spread of the virus to peripheral terminals of sensory neurons and establishment of latency, persisting lifelong within the host in ganglia innervating the head region [2,7,8].

HSV-1 has developed various strategies to invade the host by overcoming the protective barriers of the skin and mucous membranes [9,10]. It also avoids both the innate and adaptive immune systems [11,12,13] and can even block host defense mechanisms such as Toll-like receptors (TLRs) sensing, cyclic GMP–AMP synthase (cGAS) sensing, interferon response, apoptosis, endoplasmic reticulum stress response, and autophagy [12,13]. Macroautophagy (hereafter referred to as autophagy) is a highly conserved recycling and degradative pathway, which involves the biogenesis of double-membrane vesicles, called autophagosomes, which sequester cargo for subsequent degradation by hydrolytic enzymes contained in lysosomes. During this process, bulky substrates such as protein aggregates and damaged organelles are broken down and eliminated or recycled [14,15]. Currently, this process is considered a key cell-autonomous host defense mechanism that operates against pathogens such as bacteria and viruses. This process is termed xenophagy to differentiate it from the degradation of self-constituents [16,17,18]. Growing evidence shows that autophagy seems to have a dual role [19], being induced by the host to restrict viral replication and contributing to the degradation of several viruses such as Human Immunodeficiency Virus (HIV) [20,21,22], Rift Valley Fever Virus (RVFV) [23], Chikungunya virus (CHIKV) [24,25], Dengue Virus-2 (DENV-2) [25], Porcine Epidemic Diarrhea Virus (PEDV) [26], and Sindbis Virus (SINV) [27,28] or by contrast, counteracting autophagy can be a pro-viral strategy to replicate and spread such as Hepatitis B Virus (HBV) [29], Zika Virus (ZIKV) [30], Influenza A Virus [31], and Classical Swine Fever Virus (CSFV) [32,33]. 

Subversion of autophagy by HSV-1 [34,35] as a pro-viral mechanism was first reported two decades ago and involves the signaling pathway Protein Kinase R (PKR)/eIF-2α that positively regulates autophagy as a target for the viral protein known as infected cell protein 34.5 (ICP34.5), which interacts with the protein phosphatase 1 α (PP1α) and as a consequence prevents the host translational shut-off mediated by eIF-2α [36]. Furthermore, the tegument HSV-1 protein known as US11 also plays a role in hijacking this signaling pathway upstream of ICP34.5. US11 is expressed at late times during HSV-1 infection and physically interacts with PKR, preventing the activation of the PKR/eIF2α signaling pathway [37,38] and, consequently, the translational arrest, avoiding the phosphorylation of eIF2α [39] and working as an alternative mechanism to support and assure the translation of the viral proteins in host cells. It is interesting to note that autophagy can degrade HSV-1 in a PKR-dependent manner [40], while US11 can inhibit autophagy in the same manner [39]. However, in neurons and fibroblasts, during the early steps of autophagosome formation, the mechanism can be blocked by the binding of ICP34.5 with Beclin-1, a subunit associated with the Class III Phosphoinositide 3-kinase (PI3K-III) complex I [41], to form a highly regulated complex which promotes phosphatidylinositol 3-phosphate generation at autophagy initiation [42]. Moreover, in dendritic cells, ICP34.5 does not prevent the early steps of autophagy, but instead, it blocks autophagic flux towards lysosomes, leading to the accumulation of autophagosomes and the autophagic receptor Sequestosome 1 (SQSTM1/p62) [43]. This correlates with the phenotype found in HSV1-infected neuroblastoma cells [44]. Furthermore, in non-neuronal cells, the viral protein US3 with Ser/Thr kinase activity leads to the phosphorylation of autophagy regulators such as Unc-51-like kinase 1 (ULK1) and Beclin-1 [45], inhibiting its respective autophagic functions. This is another strategy used to counteract the autophagy function at an early stage of infection.

Given the biological significance of autophagy in the host–virus interaction, it is conceivable to think that HSV-1 has developed other strategies to exploit this host defense mechanism, modulating the expression and function of related factors at the autophagic machinery. A key step during the autophagic process is the recognition of the cargoes by autophagic receptors that act as a bridge between the ubiquitinated cargo and the phagophore during the elongation step of autophagy, which are subsequently degraded together with the substrates [46]. Among these, SQSTM1/p62 is accumulated and redistributed towards cytoplasmic inclusions, which correlates with autophagosomes accumulation in HSV-1-infected dendritic cell lines [43], suggesting that the autophagy is deficient in this cell type. On the other hand, research conducted on the HSV-1-infected erythroblast cell line HEL showed that during the initial stages of infection, HSV-1 reduces the levels of SQSTM1/p62 and Optineurin-1 (OPTN1) receptors, dependent on proteasome activity [47]. This indicates that HSV-1 has created various approaches to counteract the same targets depending on the type of host cell.

Even though some studies have begun to unveil the effects of HSV-1 infection on autophagic receptors, the consequences of HSV-1 productive infection on the levels of the autophagic receptors in cells physiologically permissive to HSV-1 is still unclear. Hence, our research found that xenophagy acts as a crucial cell-intrinsic host defense mechanism against HSV-1 in neuroglioma cells and keratinocytes, mediated by the clearing up of SQSTM1/p62, neighbor of BRCA1 gene 1 (NBR1), OPTN1 and nuclear dot protein 52 kDa (NDP52), key autophagic receptor proteins related to xenophagy [48]. Additionally, we found that the removal of the tegument HSV-1 protein US11 enhances the turnover of SQSTM1/p62, NBR1, OPTN1, and NDP52, in correlation to an increase in the levels of SQSTM1/p62 ubiquitinated in host cells phenotype accompanied by a reduction in progeny HSV-1 yield. Here, we found that slowing down the degradation of the autophagic receptors is another strategy employed by HSV-1 to overcome autophagy, with US11 being a potential viral factor able to restrict the process. Altogether, our findings provide new insights into the role of xenophagy in the control of HSV-1 infection by host cells and contribute new outlooks about the ways of subversion of this cellular process by HSV-1.

## 2. Materials and Methods

### 2.1. Chemical Reagents and Antibodies

The HALT protease and phosphatase inhibitor cocktail was purchased from Thermo Fisher Scientific (Waltham, MA, USA) and RIPA Lysis Buffer, Dimethyl sulfoxide (DMSO; CAS 67-68-5), Z-Leu-Leu-Leu-al (MG132; CAS 133407-82-6), Chloroquine (CQ; CAS 50-63-5), and Bafilomycin A1 (Baf A1; CAS 88899-55-2) were purchased from Sigma Aldrich (St. Louis, MO, USA).

The following monoclonal antibodies were used: mouse anti-β-actin clone BA3R (cat: MA5-15739) was acquired from Thermo Fisher Scientific, mouse anti-p62 lck ligand (SQSTM1/p62) clone 3/p62 (cat: 610833) was acquired from BD Biosciences (Franklin Lakes, NJ, USA), mouse anti-NBR1 clone 4BR (cat: sc-130380), mouse anti-Optineurin (OPTN1) clone C-1 (cat: sc-271549), anti-CALCOCO2 (NDP52) clone F-6 (cat: sc-53329), and mouse anti-ICP8 clone 10A3 (cat: sc-53329) were acquired from Santa Cruz Biotechnology (Dallas, TX, USA), mouse anti-ICP5 clone 3B6 (cat: ab6508) and mouse anti-ICP27 clone H1113 (cat: ab53480) were acquired from Abcam (Cambridge, UK), mouse anti-Ubiquitin (Ub) clone P4D1 (cat: #AUB01) was acquired from Cytoskeleton (Denver, CO, USA), anti-Myc tag clone 9B11 (cat: #2276) was acquired from Cell Signaling Technology (Danvers, MA, USA), and mouse anti-US11 and mouse anti-ICP34.5 antibodies were produced by Dr. Bernard Roizman’s Laboratory and kindly donated for this research (Northwestern University, Chicago, IL, USA). The following polyclonal antibodies were used: rabbit anti-LC3 (cat: #2775) was acquired from Cell Signaling Technology and rabbit anti-HSV-1 (cat: ab9533) was acquired from Abcam. Horseradish peroxidase-conjugated secondary antibodies anti-mouse IgG (cat: 115-035-003) and anti-rabbit IgG (cat: 111-035-003), both produced in goat, were purchased from Jackson ImmunoResearch Laboratories (West Grove, PA, USA).

### 2.2. Cell Culture

Homo sapiens brain neuroglioma cells, referred to here as H4 human neuroglioma cells (ATCC^®^ HTB-148™), and African green monkey kidney (Vero) cells (ATCC^®^ CCL-81™) were obtained from the American Type Culture Collection (Manassas, VA, USA). The human keratinocyte cell line, referred to as HaCaT cells was generously donated by Dr. Norbert Fusenig [49] (German Cancer Research Center, DKFZ, Heidelberg, Germany). Cell lines were cultured in Dulbecco’s modified Eagle’s medium (DMEM-HG; Thermo Fisher Scientific), supplemented with 10% (*vol*/*vol*) certified heat-inactivated fetal bovine serum (FBS; Cytiva, Marlborough, MA, USA) and 1% (*vol*/*vol*) Antibiotic-Antimycotic (Penicillin/Streptomycin/Amphotericin B) (Thermo Fisher Scientific), and maintained in a 5% CO_2_ atmosphere at 37 °C. Cells were grown to sub-confluence and then were HSV-1 infected or treated with drugs for further immunoblot, RT-qPCR, and immunofluorescence analyses. Assays for the detection of mycoplasma contamination in culture cells were performed periodically by PCR. For starvation assays, cells were first washed with PBS three times and incubated in Earle’s balanced salts solution (EBSS), acquired from Sigma-Aldrich (cat: E2888), for different periods of time.

### 2.3. Biosafety Methodology

All procedures involving HSV-1-propagation and infection experiments were performed in a laboratory equipped with a level 2 biosafety cabinet. All the activities were authorized and supervised by the principal investigator under institutional guidelines from the Universidad Austral de Chile. This study was approved by the Institutional Biosafety Committee of Universidad Austral de Chile (Reports N°012/20).

### 2.4. Propagation, TCID_50_ Titration, and In Vitro HSV-1 Infection

The HSV-1 strain F [50], mutant HSV-1 strain R3616 (ΔICP34.5 null) [51], and mutant HSV-1 strain R3631 (ΔUS11 null) [52,53], used in this study were kindly supplied by Dr. Bernard Roizman. The virus stocks were propagated, titrated in Vero cells [50] by TCID_50_ assay, and stored in an ultrafreezer at −80 °C for further assays. HSV-1 titer was measured through End-Point Dilution Assay known as TCID_50_ (Tissue Cultured Infective Dose), where Vero cells were plated in a 96-well plate and grown until reaching confluence. The monolayer cells were incubated for 1 h at 37 °C with serial 10-fold diluted HSV-1 suspensions in DMEM culture medium. The medium was aspirated and fresh DMEM containing 5% FBS was added, incubating the plates at 37 °C. Estimation of the end-points was made at 96 h for visualization and registry of the cytopathic effect for each dilution. The viral dose infective necessary to produce a cytopathic effect in 50% of the infected cells was calculated as TCID_50_/mL according to Spearman–Kärber method [54]. The titration was performed in three biological replicates with eight technical replicates for each dilution and condition. The final results were expressed as log_10_TCID_50_/mL.

The in vitro HSV-1 infections were carried out at a multiplicity of infection (MOI) of 10 for immunoblot and RT-qPCR experiments. The HSV-1 virions were allowed to adsorb for 1 h in a low volume of medium supplemented with 2% FBS, with regular mixing every 15 min. Following infection, the excess virus was removed by aspiration, the cells were washed twice with PBS, and finally, a fresh medium was added. Uninfected (mock) and HSV-1-infected cells were cultivated for different time periods, 4, 8, 18, and 24 h post-infection (hpi), or alternatively treated and infected.

### 2.5. RNA Isolation and RT-qPCR Analyses

Total RNA extraction from uninfected (mock), HSV-1-infected H4 human neuroglioma cells, and HaCaT HSV-1-infected cells for different time periods was carried out using the E.Z.N.A.^®^ Total RNA Kit I (Omega Bio-tek, Norcross, GA, USA) according to the manufacturer’s instructions, and either purity (260/280 nm ratio and 260/230 nm ratio) or quantity (260 nm absorbance) were determined by spectrophotometry using MaestroNano (MaestroGen, Hsinchu, Taiwan). The cDNA synthesis was performed from 2 µg of total RNA in a total volume of 15 µL using Moloney Murine Leukemia Virus (M-MLV) Reverse Transcriptase (Promega, Madison, WI, USA) and Random Primers (Promega) according to supplier’s instructions. Specific primer pairs for human genes *GADPH*, *P62* (SQSTM1/p62), *NBR1*, *OPTN* (OPTN1), and *CALCOCO2* (NDP52) and specific primers to the viral gene UL29 [55] were designed for quantitative reverse transcription PCR on cDNA template (RT-qPCR) (Appendix A). cDNA was quantified by qPCR in a total volume of 20 µL using Master mix qPCR Brilliant II Sybr ^®^ Green (Agilent Technologies, Santa Clara, CA, USA) according to supplier’s instructions in a Stratagene MX3000P Real-Time PCR Detection System (Agilent Technologies) in a 96-well plate (Biopointe Scientific, Claremont, CA, USA). In a 40-cycle PCR reaction, each cycle consisted of 10 s at 95 °C, 15 s at 55 °C, and 15 s at 72 °C. All analyses were performed in triplicate. PCR data were acquired from Software MxPro v4.10 Build 389 (Agilent Technologies), and the expression level of each gene was normalized to *GADPH* and analyzed with the 2^−ΔΔCt^ method [56]. The results were expressed as relative quantity regarding the *GADPH* normalizer [57].

### 2.6. Preparation of Protein Extracts, Electrophoresis, SDS-PAGE, and Immunoblot Analysis

For the preparation of soluble protein extracts, the cells were washed in ice-cold phosphate-buffered saline (PBS), harvested, and lysed at 4 °C in commercial RIPA buffer 1X (50 mM Tris Buffer pH 7.5, 150 mM NaCl, 5 mM EDTA, 1% *vol*/*vol* NP-40, 0.5% *wt*/*vol* sodium deoxycholate, and 0.1% *wt*/*vol* SDS) (Sigma-Aldrich) supplemented with HALT phosphatase and protease cocktail inhibitor (Thermo Fisher Scientific). All lysates were cleared by centrifugation at 16,000× *g* for 20 min at 4 °C, and protein concentration was determined with a Pierce™ BCA Protein Assay Kit (Thermo Fisher Scientific). Samples with an equal amount of proteins were boiled with SDS-PAGE sample buffer 4X (200 mM Tris Buffer pH 6.8, 8% *wt*/*vol* SDS, 40% *vol*/*vol* Glycerol, 120 mg Bromophenol blue, and 10% *vol*/*vol* β-Mercaptoethanol) for 5 min and then analyzed by SDS-PAGE. The proteins were electroblotted onto Protran^®^ immunoblotting nitrocellulose membranes (GE Healthcare, Chicago, IL, USA) and then blocked by incubation at room temperature for 30 min in PBS containing 5% (*wt*/*vol*) free-fat dry milk. Next, the samples were sequentially incubated with primary and secondary antibodies, both diluted in PBS 5% (*wt*/*vol*) BSA, for 1 h at room temperature or overnight at 4 °C. Chemiluminescence protein detection was performed using SuperSignal™ West Pico PLUS substrate (Thermo Fisher Scientific) or Westar Supernova (Cyanagen, Bologna, Italy) based on the desired level of sensitivity, and the signal was registered using iBright™ Immunoblot Imaging System (Thermo Fisher Scientific).

### 2.7. His-Ub Pulldown

To determine the ubiquitination status of SQSTM1/p62, we assessed a His-Ub pulldown under denaturing conditions [58], purifying 6His-Myc-Ub tagged proteins by Ni^2+^ affinity. H4 cells were transfected with pcDNA3-6His-Myc-Ub in 100 mm tissue culture plates using Lipofectamine 2000 transfection reagent (Thermo Fisher Scientific) according to the manufacturer’s instructions. At 12 h post-transfection, cells were HSV-1-infected with strain F or strain R3631. After HSV-1-infection, cells were harvested in ice-cold PBS. For the purification, 90% of the cell suspension was lysed in a chaotropic buffer (6 M Guanidine hydrochloride, 100 mM Sodium phosphate buffer pH 8.0, 10 mM β-Mercaptoethanol, and 20 mM Imidazole). Equal amounts of lysates were pre-cleared by rotation for 1 h at 4 °C with the uncharged His-Bind agarose resin (Novagen-Millipore, Burlington, MA, USA). After recovering, cell lysates were incubated in precharged Ni^2+^ nitrilotriacetic acid (Ni-NTA) agarose resin (Qiagen, Hilden, Germany) by rotation for 12 h at 4 °C followed by a washing protocol keeping Imidazole concentration, with variations in pH at 4 °C as follows: two times with buffer pH 8.0 (8 M Urea, 100 mM sodium phosphate buffer pH 8.0, 5 mM β-Mercaptoethanol, 20 mM Imidazole, and 0.1% (*vol*/*vol*) Triton X-100), two times with buffer pH 6.3 (8 M Urea, 100 mM sodium phosphate buffer pH 6.3, 5 mM β-Mercaptoethanol, 20 mM Imidazole, and 0.1% (*vol*/*vol*) Triton X-100), and two times with non-chaotropic buffer pH 6.8 (50 mM Tris-HCl pH 6.8, 20 mM Imidazole). The samples were eluted in SDS elution buffer (50 mM Tris-HCl pH 6.8, 450 mM Imidazole, 200 mM NaCl, 1% (*wt*/*vol*) SDS, 15% (*vol*/*vol*) glycerol, 0.72 mM β-Mercaptoethanol, and 0.01% (*wt*/*vol*) bromophenol blue) to subsequently analyze the samples by SDS-PAGE and immunoblot for total 6His-Myc-Ub conjugates using anti-Myc tag antibody. Likewise, SQSTM1/p62 conjugates were detected using anti-p62 lck ligand. Unpurified extract samples were obtained from 10% of the cell suspension and prepared for the SDS-PAGE and immunoblot for SQSTM1/p62 and β-ACTIN.

### 2.8. Bright-Field Microscopy

To evaluate cellular morphology, uninfected and HSV-1-infected cells were documented by bright-field microscopy with a CKX41SF inverted optical microscope (Olympus, Tokio, Japan) using the CAch N 10x/0.25 objective. To acquire the images, the LC20 digital camera (Olympus) was used, coupled with the microscope, and the adjustments were made with LCmicro V. 2.4 Image Analysis Software (Olympus).

### 2.9. Densitometric Quantification and Statistical Analysis

To estimate the amount of immunoblot signal, we used Image J Software version 1.54d by Wayne Rasband, NIH (https://imagej.net/ij/). To ensure adequate statistical power, we quantified protein bands from at least three independent experiments for each condition. Data analysis was performed using Microsoft Excel 2013 for Windows (Redmond, WA, USA) and GraphPad Prism 8.0. Ink. The results are represented in graphs depicting the mean ± standard deviation of the biological replicates. The statistical significance of the data from the two groups was determined with a Student’s *t*-test for parametric data. Values of *p* < 0.05 (*), *p* < 0.01 (**), *p* < 0.001 (***), and *p* < 0.0001 (****) were regarded as statistically significant and are indicated in the figures, according to the GraphPad style for *p* values. The statistical significance of data from multiple experimental groups was analyzed either using a One-Way ANOVA followed by a Tukey’s test to evaluate pair-wise comparisons, or a Two-Way ANOVA followed by Sidak’s test to compare the mean differences between groups that have been split into two independent variables. The value of *p* < 0.05 was regarded as statistically significant and was indicated in the figures.

## 3. Results

### 3.1. HSV-1 Infection Causes a Reduction in the Total Levels of Ub Conjugates and Several Autophagic Receptors in Host Cells 

HSV-1 has developed effective subversion strategies against its host by altering the expression and abundance of host proteins. To achieve such purposes, HSV-1 can induce the ubiquitination of several host proteins and their subsequent proteasome-dependent degradation, a process mediated by the immediate early viral protein E3 ligase ICP0 [59,60]. Notably, many of these targeted host proteins play roles in cellular defenses that restrict viral infection [61,62]. Considering that for the virus, to induce the ubiquitination on host proteins serves as a potent mechanism to counteract host responses, but at the same time, from the perspective of the host, ubiquitination serves as a degradative signal for xenophagy, we explored the effects of HSV-1 infection on the total levels of Ub conjugates using H4 human neuroglioma cells as a model of host cells for in vitro infections. H4 cells have been previously used due to their permissiveness for HSV-1 infection [63]. H4 cells were uninfected (mock, M) or infected with HSV-1 (strain F) at a multiplicity of infection of 10 (MOI 10) at 4, 8, 18, and 24 hpi. Subsequently, soluble extracts were analyzed by immunoblot. We used an anti-Ub antibody, which recognizes all types of Ub conjugates, including (mono) Ub and (poly) Ub conjugates (Figure 1A, first panel). Basal levels of high molecular weight Ub conjugates were observed in uninfected cells, displaying a multi-banding ladder pattern. A similar pattern was observed in HSV-1-infected cells at 4 hpi (Figure 1A, lanes 1–2). Interestingly, in HSV-1-infected cells, the levels of Ub conjugates decrease from 8 hpi onward (Figure 1A, lanes 3–5). These findings suggest that ubiquitinated proteins undergo degradation in host cells during the early stages of HSV-1 infection.

To gain insight into the cause of reduction in the total levels of Ub conjugates, under the premise that the recognition of Ub conjugates by autophagic receptors, which bind them and direct them towards autophagosomes for subsequent breakdown [48,64], could diminish the Ub conjugates, we specifically examined the endogenous protein levels of a set of autophagic receptors related to xenophagy SQSTM1/p62, OPTN1, NBR1, and NDP52 by immunoblot during HSV-1 infection in comparison to the uninfected cells (Figure 1A, second to sixth panel). Densitometric analysis revealed that the levels of the SQSTM1/p62 and OPTN1 receptors decreased significantly from 4 hpi onward when compared to uninfected cells (Figure 1A, second to fourth panel, Figure 1B,C). Interestingly, we observed a distinct migration pattern for SQSTM1/p62, suggesting it may be post-translationally modified by HSV-1 during early stages of infection, starting from 4 hpi. This is evidenced by a ladder pattern of modified species with higher molecular weight appearing above the band corresponding to SQSTM1/p62 (Figure 1A, third panel, lanes 2 and 3). This observation is clearer at higher exposures (Figure 1A, second panel, lanes 2 and 3). These modified species decrease in the later stages of infection as the SQSTM1/p62 levels reduce (Figure 1A, second and third panel, lanes 4 and 5). Similarly, during HSV-1 infection, the levels of the NBR1 and NDP52 receptors decreased significantly starting from 8 hpi, when compared to uninfected cells (Figure 1A, fifth and sixth panel, Figure 1D,E). These results suggest that HSV-1 infection in H4 cells reduces SQSTM1/p62, OPTN1, NBR1, and NDP52 levels at early stages of infection with a notable difference in its turnover kinetics. We further analyzed three representative proteins of the HSV-1 replicative cycle: ICP27 (an immediate early protein), ICP8 (an early protein), and ICP5 (a late protein, which is the major capsid protein of HSV-1). Its levels were examined in infected cells (Figure 1A, seventh to ninth panels, lanes 2–5) and compared to those uninfected cells (Figure 1A, seventh to ninth panels, lane 1) to control the efficiency of infection. The sequential detection of ICP27, ICP8, and ICP5 reveals the protein synthesis cascade of HSV-1. Moreover, we assessed the effect of viral load on H4 cells by testing different multiplicities of infection (MOI). We examined the levels of autophagic receptors SQSTM1/p62, OPTN1, NBR1, and NDP52 in HSV-1-infected cells at viral doses ranging from MOI 0.1 to 10 at 8 hpi (Appendix A, first to fifth panel) and contrasted them with uninfected cells. Immunoblot analyses showed that only high MOIs of HSV-1, specifically between MOI 5 and MOI 10, lead to a notable reduction in SQSTM1/p62, OPTN1, NBR1, and NDP52 levels in comparison to uninfected cells (Appendix A). Modified species of higher molecular weight above the band corresponding to SQSTM1/p62 were clearly observable at higher exposures between MOI 1 and MOI 10 (Appendix A, first panel, lanes 3 to 5), consistent with Figure 1A. Furthermore, we analyzed ICP27, ICP8, and ICP5 levels to validate the in vitro infection (Appendix A, sixth to eighth panels). Notably, all the proteins of the expression cascade of HSV-1 were only detectable between MOI 5 and 10, which is in line with the observed reduction in the autophagic receptors.

HSV-1 establishes a productive infection in the host, showing a high biological affinity with epidermal keratinocytes from the oral mucosa, skin, or cornea during the primoinfection [65]. Due to this, keratinocytes represent one of the most relevant cell types for infection studies [65,66]. We utilized HaCaT epidermal keratinocytes as another cellular model for HSV-1 productive infection. Total levels of Ub conjugates are presented in Figure 2 (first panel). Additionally, we assessed the same set of autophagic receptors, as previously presented in H4 cells, using immunoblot analyses (Figure 2, second to fifth panels). Levels of ICP27, ICP8, and ICP5 were also analyzed to control the efficiency of infection (Figure 2 sixth to eighth panels, lanes 2–5), and compared to uninfected cells (Figure 2 sixth to eighth panels, lane 1). A pronounced multi-banding ladder pattern was observed in uninfected cells (Figure 2A, first panel, lane 1). This pattern became more intense in HSV-1-infected cells at 4 and 8 hpi (Figure 2A, first panel, lanes 2 and 3) but declined sharply from 18 hpi and onward (Figure 2A, first panel, lanes 4 and 5). Densitometric analyses revealed that the endogenous levels of all autophagic receptors tested decreased during HSV-1 infection compared to uninfected cells, but with distinct decline kinetics (Figure 2B–E). Specifically, the levels of OPTN1 reduced significantly, but in a transitory manner, at 4 hpi and 8 hpi, with a recovery in the levels similar to the mock condition in late stages of infection (Figure 2A, third panel, and Figure 2C). NBR1 levels significantly decrease from 4 hpi and onward (Figure 2A, fourth panel, and Figure 2D). Meanwhile, SQSTM1/p62 and NDP52 showed a significant reduction starting from 8 hpi onward (Figure 2A, second and fifth panels, lanes 2–5, Figure 2B,E), compared to uninfected cells. The viral proteins analysis demonstrated the sequential expression of ICP27, (Figure 2A, sixth panel), ICP8 (Figure 2A, seventh panel) and ICP5 as the infection progressed (Figure 2A, eighth panel). All these proteins were detectable from 4 hpi (Figure 2A, lanes 2–5), contrasting with uninfected cells (Figure 2A, lane 1) which showed no expression of HSV-1 markers, thereby confirming the permissiveness of HaCaT cells to HSV-1 infection. 

Taken together, these results clearly show that all examined autophagic receptors; SQSTM1/p62, NBR1, NDP52, and OPTN1 undergo a decline in protein levels during HSV-1 infection in H4 and HaCaT permissive cell lines. Notably, the impact of HSV-1 infection was most marked on the levels of NBR1 and NDP52. Moreover, these findings correlate with the reduction in total levels of Ub conjugates in both host cell lines.

Given that HSV-1 encodes an endoribonuclease that is responsible for the shutoff of host proteins [67], we sought to investigate the mechanism underlying the reduction in protein levels of the autophagic receptors during HSV-1 infection. To this end, we measured the mRNA levels of these receptors. Consequently, both uninfected (mock) and HSV-1-infected HaCaT and H4 cells (strain F, at MOI 10) were analyzed at 4, 8, 18, and 24 hpi using quantitative RT-PCR (RT-qPCR). The target mRNA amount in each condition was normalized to the mRNA expression of *GADPH*. Relative quantifications of mRNA expression for autophagic receptors are depicted for HaCaT cells (Appendix A) and H4 cells (Appendix A). Relative mRNA levels of both *P62* (SQSTM1/p62) (Appendix A) and *OPTN* (OPTN1) (Appendix A) are significantly diminished during late stages of HSV-1 infection compared to uninfected cells. Intriguingly, relative mRNA levels of both *NBR1* (Appendix A) and *CALCOCO2* (NDP52) (Appendix A) significantly increased at the late infection time point, specifically at 18 hpi. This suggests that HSV-1 infection can influence the transcriptional regulation in the tested autophagic receptors but with important differences between them. At this level of regulation, mRNA increase observed for *NBR1* and *CALCOCO2* at late-stages of HSV-1 infection may suggest a compensatory response to the significant reduction in NBR1 and NDP52 protein levels. Nevertheless, while these findings do not fully explain the overall reduction in the analyzed proteins, strongly suggest that the observed alterations might be driven by cellular post-translational mechanisms that regulate intracellular protein abundance. 

### 3.2. Reduction in the Levels of LC3 during HSV-1 Infection Is Enhanced in EBSS-Starved Host Cells

Given that HSV-1 induces a decrease in levels of autophagic receptors related to xenophagy, which could correlate with the activation of this process, and due to the fact that host cells often activate such a defensive mechanism against intracellular pathogens [48], we investigated the regulation of this cellular pathway during HSV-1 infection. We decided to analyze the levels of a specific marker for monitoring autophagy in mammals, microtubule-associated protein 1A/1B light chain 3 (LC3). Upon initiation of autophagy, cytosolic LC3 (hereafter referred to as LC3-I) becomes conjugated with phosphatidylethanolamine and associates with pre-autophagosomal membranes generating LC3-II [68,69]. For this, we evaluated the levels of the autophagosomal marker LC3 by immunoblot in H4 cells during HSV-1 infection (Figure 3A). Densitometric analyses revealed a significant reduction in the endogenous levels of LC3-I from 8 hpi (Figure 3B) and LC3-II from 4 hpi and onward when compared to uninfected cells (Figure 3C). The simultaneous reduction in both LC3-I and LC3-II levels, along with the decreased levels of autophagic receptors suggests that HSV-1 infection could be triggering autophagy in host cells as a potential defensive response against the virus.

To understand the significance of reduced LC3 levels, realizing that the host as well as HSV-1 can modulate the autophagy pathway, we evaluated whether a lack of nutrients, a common condition to induce autophagy [68,70], could enhance the autophagy during HSV-1 infection, under the assumption that the host could promote this process. Following this line of thought, whether this stimulus contributes to the reduction in HSV-1 progeny yield, this would mean that it is a host defense mechanism against HSV-1. To test this, we treated H4 cells with Earle’s balanced salts solution (EBSS), a medium devoid of essential amino acids, which is a well-established condition to induce autophagy [68,70]. First, to determine the optimal conditions for the activation of autophagy, we tested the effect of EBSS on the endogenous levels of LC3-I and LC3-II in H4 cells in a time course between 30 min and 8 h (Appendix A, lanes 2 to 6). The densitometric quantification of LC3-I showed that the levels of the autophagosomal marker began to decrease significantly after 8 h EBSS (Appendix A), indicating increased activity of the autophagy pathway. Similarly, we observed that levels of LC3-II decreased significantly after 4 h of EBSS (Appendix A). Hence, a significant decline in both LC3-I and LC3-II was found after 8 h of treatment (Appendix A) compared to control cells (Appendix A, lane 1). Pharmacological treatment with 0.1 mM Chloroquine (CQ) for 4 h was used as a control for LC3-II accumulation (Appendix A, lane 7). CQ is a weak base that diffuses into the lysosomes and undergoes protonation, impairing autophagic degradation in lysosomes, and is widely used as an autophagic flux inhibitor [66]. As shown in Appendix A, the densitometric quantification of LC3-I showed that the levels of the autophagosomal marker began to decrease significantly after 8 h EBSS (Appendix A), indicating increased activity of the autophagy pathway. Similarly, we observed that levels of LC3-II decreased significantly after 4 h of EBSS, as shown in Appendix A. Consistent with inhibition of autophagic flux, CQ treatment caused a significant reduction in LC3-I (Appendix A, lane 7 and Appendix A), besides a significant increase in LC3-II (Appendix A, lane 7 and Appendix A). Thus, the effect of HSV-1 infection under these metabolic conditions was evaluated, in H4 cells grown in EBSS for 4 and 8 h, in the absence (Figure 3D, lanes 1 to 3) and presence of HSV-1 at MOI 10 (Figure 3D, lanes 4 to 6), using as a control, cells incubated in fully supplemented culture medium in both cases. After 8 hpi, soluble extracts were subjected to immunoblot, and the endogenous levels of LC3-I and LC3-II were evaluated (Figure 3D–F). Densitometric analyses revealed that in uninfected cells, EBSS caused a significant decrease in LC3-I levels from 4 h and onward in comparison to the control condition (Figure 3D, lanes 1 to 3; Figure 3E, bars 1 to 3). Similarly, LC3-II levels were significantly decreased from 4 h of EBSS and onward (Figure 3D, lanes 1 to 3; Figure 3F, bars 1 to 3). Moreover, 8 h of treatment with EBSS showed similar levels of LC3-I in both uninfected and infected cells (Figure 3D, lane 3 and 6; Figure 3E, bars 3 and 6). In contrast, EBSS caused a significant decrease in the levels of LC3-II on infected cells compared to uninfected cells (Figure 3D, lanes 3 and 6; Figure 3F, bars 3 and 6). In addition, to determine whether the balance of LC3-I and LC3-II reflects the effect of the HSV-1 on the EBSS stimuli, we investigated the response of the cytosolic LC3 ratio [71] under these metabolic conditions, in uninfected and HSV-1-infected cells, as an index of conversion of LC3-I to LC3-II. The LC3-II/LC3I ratio in each condition was calculated from the values obtained in Figure 3D (Appendix A). The ratio showed reduced levels of LC3-II concomitant to reduced levels of LC3-I on infected cells compared to uninfected cells (Appendix A, bars 1 and 4) following the downward trend at 4 h (Appendix A, bars 2 and 5) and 8 h (Appendix A, bars 3 and 6) of treatment with EBSS on infected cells compared to uninfected cells. These findings indicate that the transformation of LC3-I to LC3-II is increased during HSV-1 infection either in basal or starved conditions.

These results strongly indicate that during HSV-1 infection, autophagy is an enhanced process that is boosted by EBSS. This enhancement of autophagy, combined with the reduction in specific autophagic receptors, suggests that the host activates a more specific form of autophagy, which is likely xenophagy leading to degradation of LC3.

### 3.3. Reduction in the Levels of Autophagic Receptors during HSV-1 Infection Is Enhanced in EBSS-Starved Host Cells

Given that autophagic receptors sequester ubiquitinated cargoes into autophagosomes and are degraded along the cargoes, we investigated whether EBSS treatment might enhance the reduction in SQSTM1/p62, OPTN1, NBR1, and NDP52 autophagic receptors levels during HSV-1 infection. We evaluated this by immunoblot in H4-infected cells under starvation conditions, comparing the results to those of uninfected starved cells (Figure 4). Densitometric analyses showed that the levels of SQSTM1/p62, OPTN1, NBR1, and NDP52 in uninfected cells significantly decreased at 8 h of EBSS, compared to cells treated with fully supplemented medium (Figure 4A, lane 3 compared to lane 1 and Figure 4B–E, bars 1 and 3). However, the receptors exhibited varying kinetics of decline in response to EBSS in H4 cells. In line with the findings shown in Figure 1, the levels of all tested autophagic receptors were markedly lower in H4 HSV-1-infected cells compared to uninfected cells (Figure 4A, lane 4 compared to lane 1 and Figure 4B–E, bars 1 and 4). Moreover, after 8h of EBSS treatment, the levels of OPTN1, NBR1, and NDP52 in HSV-1-infected cells showed a significant decline compared to uninfected cells treated with 8 h of EBSS (Figure 4A, lane 6 compared to lane 3 and Figure 4C–E, bar 3 and 6). However, similar levels of SQSTM1/p62 were observed in infected cells treated with EBSS compared to uninfected cells treated with EBSS (Figure 4A, lane 6 compared to lane 3 and Figure 4B, bar 3 and 6). These findings suggest that in HSV-1-infected cells, OPTN1, NBR1, and NDP52 levels decrease even further in response to nutrient deprivation. Of all receptors tested, NBR1 displays the most pronounced phenotype compared to OPTN1 and NDP52—a distinction that, interestingly, is not observed with SQSTM1/p62. Together, these findings strengthen the notion that the host activates xenophagy during HSV-1, a process that could be enhanced during lack of nutrients, leading to degradation of autophagic receptors.

### 3.4. Starvation Enhances the Negative Impacts of Xenophagy on the HSV-1 Replication

Autophagy plays a dual role, acting either as an intrinsic host defense against viruses or being exploited by viruses to enhance their replication in host cells [19]. Following the notion that the host could promote autophagy during HSV-1 infection, we evaluated whether the lack of nutrients contributes to a reduction in HSV-1 progeny yield. Considering this, we aim to determine the impact of starvation-induced autophagy on the expression of HSV-1 proteins and progeny yield of HSV-1 using EBSS. To achieve this, both uninfected and HSV-1 infected H4 cells were treated with EBSS for 4 and 8 h (Figure 5A), with fully supplemented culture medium serving as a control condition in both scenarios. After 8 hpi, the soluble extracts were analyzed through immunoblot to examine the three representative viral proteins indicative of the HSV-1 replicative cycle, ICP27, ICP8, and ICP5, along with multiple HSV-1 proteins (Figure 5A, lanes 4–6). As expected, no viral proteins were detected in uninfected cells (Figure 5A, lanes 1–3). Interestingly, EBSS led to a decrease in the expression of several HSV-1 proteins (Figure 5A–D). Densitometric analyses supported these observations, revealing a significant decrease in all tested viral proteins after 8 h of EBSS treatment compared to the control (Figure 5B–D). Therefore, these findings suggest that the activation of autophagy via starvation acts as an intrinsic host defense mechanism against HSV-1, likely by facilitating its degradation in an autophagy receptor-dependent manner. According to our hypothesis, we set out to determine whether autophagy could act as a cytoprotective pathway against HSV-1 infection by examining the influence of EBSS on progeny HSV-1 yield. To accomplish this, we compared the production of HSV-1 under nutrient-deprived conditions using EBSS in contrast to standard growth conditions with a fully supplemented culture medium. We collected cell culture supernatants at 4, 8, 18, and 24 hpi and assessed the production of progeny virus using TCID_50_ titration in Vero cells. Notably, starting from 8 hpi, the viral titer of cells treated with EBSS was significantly reduced by roughly two orders of magnitude compared to those from cells in a fully supplemented culture medium (Figure 5E). These results indicate that starvation-induced autophagy prevents the replicative cycle of HSV-1. Taken together, our findings underscore the role of xenophagy as an intrinsic host defense mechanism that counteract HSV-1 replication, with detrimental effects on both the expression of intracellular viral proteins and the release of progeny HSV-1 from the host.

### 3.5. The Degradation of Autophagic Receptors in HSV-1-Infected Cells Depends on the Autophagic Flux

Several studies have shown that Ub receptors link the autophagy pathway and the ubiquitin-proteasome system [72,73]. Moreover, it has been reported that SQSTM1/p62 and OPTN1 are targeted by the viral E3 Ub ligase ICP0 during the early stages of HSV1 infection, driving them towards the proteasome [47]. Given these insights, and that our results reveal that autophagy regulates these receptors during HSV-1 infection, to determine whether the activity of autophagic flux impacts the autophagic receptor levels during HSV-1 infection, we investigated the effects of lysosomal pharmacological inhibition treating cells with Bafilomycin A1 (Baf A1), which is widely used in vitro as an autophagic flux inhibitor. This drug is a specific inhibitor of the vacuolar-type H^+^-ATPase pump (V-ATPase) in cells. This action prevents lysosomal acidification, leading to an increase in lysosomal pH, thereby disrupting autophagosome–lysosome fusion and effectively blocking the autophagic flux [74]. H4 cells were incubated with Baf A1 for 8 h in the absence and presence of HSV-1, using DMSO as a control condition. Soluble protein extracts were subjected to immunoblot to evaluate the levels of SQSTM1/p62, OPTN1, NBR1, and NDP52 (Figure 6A). Densitometric analysis of the immunoblot showed Baf A1 treatment in uninfected cells caused no significant changes in SQSTM1/p62 levels (Figure 6B) but did cause a significant increase in the levels of OPTN1, NBR1, and NDP52 receptors, compared to untreated cells (Figure 6A,D,F,H). On the other side, in HSV-1-infected cells, Baf A1 caused a significant increase in all receptors tested (Figure 6A,C,E,G,I). Altogether, these findings strongly indicate that the degradation of autophagic receptors in HSV-1-infected cells depends on the autophagic flux. Because, curiously, it has been reported that specific autophagic receptors may also be led to the proteasome during HSV-1 infection [47], we also tested the contribution of the proteasome in the degradation of the autophagic receptors. For this, we tested the effect of proteasome inhibition using 20 μM MG132 for 8 h in uninfected and HSV-1-infected cells (Appendix A). Densitometric analyses showed no significant changes in the levels of SQSTM1/p62, OPTN1, NBR1, and NDP52 receptors in uninfected cells upon MG132 treatment, compared to control cells (Appendix A). However, in infected cells, MG132 treatment caused a significant increase in the levels of NDP52, while none of the other receptors showed an increase under similar conditions (Appendix A). Moreover, we observed a decrease in SQSTM1/p62 levels in HSV-1-infected cells following MG132 treatment (Appendix A), a decrease that could be attributed to autophagy triggered by the cellular stress caused by both MG132 and the viral infection. Altogether, these results reveal that during HSV-1 infection, the lysosomal function of host cells contributes to the decrease in SQSTM1/p62, OPTN1, NBR1, and NDP52 autophagy receptors. Moreover, our findings strongly indicate that during HSV-1 infection, the levels of NDP52 are controlled by both degradation systems. This suggests that during infection host cells could activate cooperative degradative systems as a powerful cell-intrinsic host defense mechanism. 

### 3.6. The Tegument HSV-1 Protein US11 Slows Down the Decline of the Autophagic Receptors during HSV-1 Infection

It has been reported that HSV-1 employs multiple mechanisms to counteract autophagy in host cells, mediated by the viral proteins ICP34.5 [36,41,43,44] and US11 [37,38,39]. Until now, it is known that HSV-1 through ICP34.5 prevents the formation of the phagophore by binding to Beclin-1 [41] and hijacking the signaling axis of PKR/eIF2α [36]. Conversely, US11 also targets the PKR signaling, thereby preventing the shut-off of protein translation in host cells [39]. While certain functions of these viral proteins in counteracting autophagy are known, a comprehensive understanding of its roles in regulating autophagy during HSV-1 infection remains to be documented. Here, we investigated whether these two viral genes could be involved in the reduction of the levels of the autophagic receptors. To assess the impact of ICP34.5 and US11 on the regulation of the levels of the autophagic receptors, we utilized HSV-1 mutant strains, R3616 null mutant for ICP34.5 (Δ34.5) and R3631 null mutant for US11 (ΔUS11). We conducted in vitro infections using both strains and compared them to the HSV-1 strain F (wild-type; WT). Mutants were generated by homologous recombination between DNA from parental HSV-1 strain F and DNA previously mutagenized [51,52,53]. H4 cells were infected with HSV-1 strain F and the mutant versions R3616 and R3631 for 8 hpi at MOI 10. Then, with the soluble extracts, the levels of p62/SQSTM1, OPTN1, NBR1, and NDP52 were evaluated by immunoblot (Figure 7A). Likewise, it was checked that the mutant versions used for in vitro infections had suffered deletions of genes encoding ICP34.5 and US11, respectively. With the soluble extracts, ICP34.5 and US11 were evaluated by immunoblot in all the versions used in the assay (Figure 7F). From densitometric analysis of autophagic receptors, we observed that all HSV-1 versions studied significantly reduced the levels of SQSTM1/p62 (Figure 7B), OPTN1 (Figure 7C), NBR1 (Figure 7D), and NDP52 (Figure 7E) compared to uninfected cells. When we compare the effect of the in vitro infection of HSV-1 strain R3616 (Δ34.5) with HSV-1 strain F, we found no significant differences in the levels of the autophagic receptors tested (Figure 7A, lane 3 compared to lane 2, and Figure 7B–E). Interestingly, we observed that strain R3631 (ΔUS11) led to a significant reduction in the levels of all analyzed autophagic receptors compared to those infected with strain F (Figure 7A, lane 4 compared to lane 2, and Figure 7B–E). Moreover, this effect was specific for the autophagic receptors since in HSV-1 strain R3631 (ΔUS11) infected cells there were no significant changes in the protein levels of LC3-I (Appendix A) or LC3-II (Appendix A) compared to HSV-1 strain F infected cells. Thus, these results suggest that the presence of tegument HSV-1 protein US11 is involved in the regulation of autophagic receptors in host cells counteracting the degradation of the autophagic receptors during HSV-1 infection, highlighting the role of US11 as a viral regulator that operates as a brake on this process. Once again, these findings underscore the ability of HSV-1 to manipulate host factors with precision to support its replication, positioning US11 as a potential viral regulator of autophagy in host cells.

### 3.7. The Tegument HSV-1 Protein US11 Is Involved in the Ubiquitination of SQSTM1/p62 

Our findings suggest that SQSTM1/p62 might undergo some type of post-translational modification (PTM) during the early stages of HSV-1 infection, supported by a multi-banding ladder pattern which is detected to a high molecular weight above the band corresponding to SQSTM1/p62 (Figure 1A, second panel), that was even detected in HSV-1-infected cells with mutant strains R3616 and R3631 (Figure 7A, first panel), and that apparently decline in HSV-1 infected EBSS-starved cells (Figure 4A, first panel). Given that ubiquitination represents a type of PTM initiating degradative processes, including selective autophagy [75], we evaluated this possibility by performing assays with SQSTM1/p62, further evaluating the potential involvement of the viral protein US11. For this purpose, we performed a His-Ub pulldown under highly denaturing conditions; an assay that specifically enriches proteins covalently modified with ubiquitin. First, H4 cells were transfected with 6His-Myc-Ubiquitin for 12 h. Post-transfection, the cells were infected either with HSV-1 strain F or HSV-1 strain R3631 (ΔUS11) at MOI 10 for 8 hpi. The total Ub-His-Myc-conjugated proteins were purified by affinity using a Ni^+2^-loaded resin under denaturing conditions in a 6 M guanidine hydrochloride solution. Subsequently, the purified fractions were analyzed by immunoblot (IB: Ni^+2^ PULLDOWN) (Figure 8B) and compared with soluble extracts obtained from detergent-homogenized cells in RIPA buffer (IB: INPUT) (Figure 8A). In the fractions derived from detergent-homogenized cells, we observed that the total levels of SQSTM1/p62 were reduced in cells infected with the F and R3631 (ΔUS11) strains compared to uninfected cells (as depicted in Figure 8A, first panel, lanes 1–3), which undergo a sharper decline in the cells infected with strain R3631 (ΔUS11) than strain F (Figure 8A, first panel, lanes 2 and 3), in accordance with the results from Figure 7A. Interestingly, in the Ni^+2^-purified fractions there was an increased pattern of total ubiquitinated proteins in cells infected with both F and R3631 strains compared to uninfected cells (Figure 8B, lanes 1–3). Consistently, we noticed high molecular weight ubiquitinated species of SQSTM1/p62 above 63 kDa in uninfected cells (Figure 8B, lane 4) that increases in the cells infected with both strains (Figure 8B, lanes 5 and 6). Interestingly, we detected a sharper pattern of ubiquitinated species of SQSTM1/p62 in the cells infected with strain R3631 than in strain F (Figure 8B, lanes 5 and 6). Altogether, in agreement with previous findings, these results indicate that HSV-1 infection enhances the ubiquitination of SQSTM1/p62 for its degradation via autophagy. Furthermore, the results indicate that the HSV-1 protein US11 may negatively modulate this PTM on SQSTM1/p62.

### 3.8. The Removal of Tegument HSV-1 Protein US11 Impairs HSV-1 Replication

Because the roles of the viral proteins ICP34.5 and US11 in the virus–host interaction still need to be better addressed, we finally explored the influence of its removal in both intracellular viral protein levels and on HSV-1 progeny yield. To carry this out, H4 cells were infected with HSV-1 strain F and the mutant versions R3616 (ΔICP34.5) and R3631 (ΔUS11) for 8 hpi at MOI 10. Then, the soluble extracts were analyzed through immunoblot to examine several viral proteins as was conducted in Figure 6A (Figure 9A). From densitometric analysis of HSV-1 proteins, we observed that both mutant HSV-1 strains significantly increased the levels of ICP27 (Figure 9B), ICP8 (Figure 9C), and ICP5 (Figure 9D) compared to cells infected with HSV-1 strain F. When we compared the effect of the infection of HSV-1 strain R3616 (ΔICP34.5) with HSV-1 strain R3631 (ΔUS11), we found no significant differences in the levels of the viral proteins ICP27 and ICP8 (Figure 9B,C) but we found a significant increase in the levels of ICP5 (Figure 9D). Interestingly, from the analysis of HSV-1 total proteins (Figure 9A, fourth panel) we noticed a notable increase in both mutant HSV-1 strains compared to cells infected with strain F (Figure 9A, fourth panel, lanes 1–3). However, we detected a sharp increase in the HSV-1 total proteins in cells infected with strain R3631 compared to cells infected with strain 3616 (Figure 9A, fourth panel, lanes 2 and 3). Overall, we compared the production of HSV-1 progeny yield between the strains studied. We collected cell culture supernatants at 8 hpi and assessed the production of progeny virus using TCID_50_ titration in Vero cells. Notably, at 8 hpi, the viral titers of cells infected with both mutant strains were significantly reduced by roughly three orders of magnitude compared to cells infected with strain F (Figure 9E) and without significant differences between them (Figure 9E, bars 2 and 3). 

HSV-1 is a cytolytic virus whose replicative cycle ultimately leads to destruction in culture cells. The cytopathic effect (CPE) of HSV-1 infection is characterized by an early rounding up of cells upon infection and becomes more severe with increasing times of infection or viral doses [76]. In order to evaluate whether the phenotypes found in HSV-1 mutant strains can influence the CPE of HSV-1 on infected cells, we acquired images of the cell morphology by bright-field microscopy. At 8 hpi, the corresponding uninfected cells were observed to be a confluent cell monolayer (Appendix A, upper left image), and they presented no sign of CPE. Cells infected with HSV-1 strain F could hardly be distinguished from uninfected cells since several infected cells displayed as rounded, and they tended to lose cell–cell contacts (Appendix A, upper right image). Interestingly, the cell morphologies observed after infection with either HSV-1 strain R3616 (ΔICP34.5) or R3631 (ΔUS11) were essentially identical (Appendix A, lower images). This observation is consistent with the fact that both strains produced infective virions to a similar extent, as seen by TCID_50_ titration.

These findings strongly suggest that US11 could have a remarkable impact on maintaining an adequate balance between the levels of intracellular viral proteins and the egress of infectious particles from host cells. In this line of thought, the lack of US11 shifts the balance toward a greater viral protein loading within the host cells, with respect to the lack of ICP34.5, which is subject to be degraded by xenophagy, mediated by autophagic receptors, as is consistent with our previous findings. Consequently, these facts negatively impact the progeny HSV-1 yield.

## 4. Discussion

The intrinsic host defense mechanisms against viruses and bacteria, also referred to as cell-autonomous defense, can be displayed by all cell types to distinguish it from the innate immune response orchestrated by specialized cells [77]. To gain access into the host, HSV-1 has multiple strategies to overcome human barriers, including the intrinsic host defense mechanisms and the immune system [11,12], presumably because viral determinants have been selected during the co-existence of HSV-1 with humans since ancient times, to successfully evade the host responses [78]. These abilities define the “virus–host relationship” of HSV-1–human, as a virus that establishes a long-term infection. Then, we can assume that the virus–host interaction is an old and constant battle between them, in which both have developed molecular mechanisms to counteract the effect of the other. Consequently, growing evidence sustains that, as a virulence mechanism, HSV-1 has developed strategies to counteract the autophagy function [34,35]. Autophagy is a conserved homeostatic mechanism to recycle and degrade cellular components [14,15], is one of the main cellular routes responsible for maintaining cellular proteostasis, and it serves as a powerful host defense mechanism to limit virus replication in a process known as xenophagy [16,17,18]. The strategies of HSV-1 to evade autophagy have been described and involve inhibition of early steps of autophagosomes formation [41,45] and inhibition of autophagy flux [43,44]. Other indirect mechanisms involve a considerable extent of hijacking of host factors from autophagy-related routes, such as PKR/eIF-2α and TRIM23/TBK1 signaling pathways, to support their replication [36,37,38,39,79]. However, despite consistent evidence, pro-viral strategies to evade autophagy seem to depend on the cell type employed as the host. For this reason, two different cell lines were employed in this study, representative of neurons and keratinocytes, just as it happens during HSV-1 primoinfection in the natural host.

In early stages of infection, we have reported that HSV-1 triggers an early reduction in the total levels of Ub conjugates in host cells, indicative of a massive degradation induced during HSV-1 infection. During the investigation of the cellular mechanisms involved in this phenomenon, an outstanding finding was that the levels of proteins SQSTM1/p62, OPTN1, NBR1, and NDP52, the major selective autophagic receptors involved in xenophagy [17], were crosswise reduced during HSV-1 infection in human neuroglioma cells (H4); the outcome was subsequently replicated in human keratinocytes (HaCaT). A common characteristic of these cell lines is that HSV-1 does display a fast replication, and permissiveness was already tested in previous reports which included HSV-1 in vitro infections [63,65,66]. Our findings demonstrated that autophagy is activated during HSV-1 infection in H4 cells; moreover, it is the main route to degrade selective autophagic receptors in the infectious context. These results give further evidence of the role of autophagy in regulating HSV-1 infection.

The way in which autophagy contributes to the antiviral host response against HSV-1 is still controversial because our results reveal that the host can activate autophagy to eliminate the virus, a process that is enhanced in starved cells, which is opposite to some previous reports that convey autophagy as a pathway sequestered by HSV-1. These findings were confirmed by monitoring the autophagy process by measuring the endogenous levels of LC3-I and LC3-II. Moreover, our findings show that the lack of nutrients enhances autophagy, with hostile consequences for both the production of HSV-1 intracellular proteins and the growth of HSV-1 in vitro. These strong pieces of evidence attest to the importance of autophagy as a host defense mechanism against HSV-1. In this regard, this is the first report with solid biochemical evidence that supports the idea that the replication of HSV-1 is frustrated by autophagy, which is in agreement with previous morphological findings [40]. Tallóczy et al. reported by ultrastructural analyses that xenophagy plays a role in degrading HSV-1 particles in primary cultures of MEFs and mouse sympathetic neurons, based on the presence of virions visualized inside autophagosomes by electron microscopy, as well as an accelerated degradation rate of viral proteins after infection with mutant HSV-1 strain ΔICP34.5 with intact PKR function [40]. Interestingly, in H4 cells, the mutant HSV-1 strain ΔICP34.5 had no effect on the autophagy process, or at least over the autophagy-related factors studied (selective autophagy receptors and LC3 levels), compared to HSV-1 strain F. Therefore, the possibility that the viral protein ICP34.5 may affect other signaling pathways cannot be ignored. This is because an apparent decrease was spotted in the levels of the viral protein US11 in H4 cells that were infected with HSV-1 strain ΔICP34.5 as shown in Figure 7F. Here, it was reported that structural (ICP5) and non-structural (ICP27 and ICP8) HSV-1 proteins could be cargoes for degradation by autophagy, which may not necessarily correlate with the degradation of entire HSV-1 particles. However, in support of this idea, our findings revealed that autophagy leads to a drastic restriction in the HSV-1 yield in vitro under autophagy induction by nutrient starvation. In agreement with our results, it was demonstrated that inducing autophagy by lack of nutrients in MEFs and epithelial cell lines can limit HSV-1 infection in vitro, without affecting cell viability [80], which is indicative of the cytoprotective role of autophagy in host cells. Surprisingly, in MEFs derived from ATG5 KO mice, the lack of autophagy appears not to affect the replication of the mutant HSV-1 strain ΔICP34.5 [81]. Despite discrepant findings, it is plausible to speculate that alternative, xenophagy-independent cellular mechanisms might be involved in HSV-1 replication.

On the other hand, the way in which the selective autophagy receptors are controlled during HSV-1 infection is a poorly explored topic. Autophagy receptors act as mediators between cargoes and autophagosomal membranes, being essential factors for the selection and recruitment of ubiquitinated proteins towards autophagosomes, ensuring its degradation and, potentially, also the elimination of viruses [17]. According to several authors, SQSTM1/p62 forms filaments that capture and present ubiquitinated cargoes for recognition by the autophagic system [82]. On the other hand, NBR1 is also involved in the formation of ubiquitin condensates and plays a role in the initiation stage of autophagy [83]. However, viruses such as HSV-1 have evolved to evade barrier hosts targeting the autophagic receptors, as we report here. According to our findings, the host promotes the clearance of SQSTM1/p62, OPTN1, NBR1, and NDP52 by autophagy during HSV-1 infection. Additionally, the lysosomal pathway is a route that contributes to manage the levels of these host factors, as demonstrated. Interestingly, a study carried out in HEL cells revealed that HSV-1 reduces the levels of SQSTM1/p62 and OPTN1, dependent on proteasome activity during the early steps of infection [47]. In addition, the ectopic expression of SQSTM1/p62 causes restriction of the HSV-1 yield in vitro, which was enhanced in the presence of an autophagy inducer [47]. Altogether, these findings lead us to think that these host factors play a role in host viral susceptibility to HSV-1, probably playing a role in the recognition and recruitment of cargoes during HSV-1 infection or even leading to degradation of viral particles thus contributing to host intrinsic response against viral invasion.

On the other hand, our results revealed an unexpected role of the HSV-1 protein US11 over host cellular homeostasis. US11 is a non-structural protein, required for translation regulation, expressed late in the viral replicative cycle [84]. From the accumulated evidence, US11 seems to be a key HSV-1 tool that sequesters both intrinsic host defense mechanisms and innate immune response, which thwart infections early; there is strong evidence for its roles in hijacking apoptosis, IFN production, and autophagy. US11 is a multifunctional protein that has been shown to protect HeLa cells from induced apoptosis [85]. Even more recently, it was reported that US11 could recruit caspase-8 and induces cleavage, which does not trigger apoptotic signaling [86]. On the other hand, US11 binds to RIG-I and MDA-5, two cellular antiviral proteins involved in viral dsRNA recognition, and inhibits their downstream signaling pathway, preventing the production of IFN-β, a key antiviral response of the host immune system [87]. 

In the early steps of HSV-1 infection, US11 binds dsRNA and physically associates with PKR [37], avoiding PKR activation and in consequence precludes activation of the PKR/eIF2α signaling pathway [38,88], counteracting the host translation shut-off. Consequently, the interaction of US11 with PKR blocks the autophagy response in host cells [39]. The ectopic expression of US11 can block autophagy and autophagosome formation in both HeLa cells and fibroblasts [39]. However, in this research, H4 cells infected with HSV-1 strain ΔUS11 were no different in the levels of LC3 compared to cells infected with HSV-1 strain F. However, it is indeed very striking that HSV-1 has acquired multiple strategies to target the same pathway. In this study, we found that SQSTM1/p62, OPTN1, NBR1, and NDP52 are novel host factors targeted for HSV-1 protein US11. Moreover, in this investigation we provided evidence of a novel strategy of US11 to evade the host, modulating the ubiquitination of SQSTM1/p62. The removal of US11 protein from HSV-1 facilitates the ubiquitination, indicating that US11 slows down the autophagy turnover of SQSTM1/p62, providing new insights about the ways of subversion of autophagy by HSV-1. 

SQSTM1/p62 activity is tightly regulated by phosphorylation, acetylation, and ubiquitination. These modifications can modulate its capacity to recruit substrates [89,90] as well as its spatial organization [91]. Accordingly, it would be revealing in future studies to characterize the specific domains and residues that are ubiquitinated in SQSTM1/p62 under HSV-1 infection. 

It is worth mentioning the participation of viral proteins in the regulation of host ubiquitination. HSV-1 through ICP0 protein, with E3 ubiquitin ligase activity, plays a crucial role in hijacking the host cell ubiquitination machinery, which allows it to target several host cell proteins, including autophagic receptors, to create an environment permissive for virus replication [62]. On the other hand, US11 has also been implicated in interfering with autophagy through the disassembly of the TRIM23/TBK1 complex [79], which mediates virus-induced autophagy [92]. In this signaling pathway, TANK-binding kinase 1 (TBK1) mediates type I interferon induction as an early host defense mechanism and notably facilitates the recruitment of SQSTM1/p62 via direct interaction and phosphorylation [88]. Interestingly, the formation of the SQSTM1/p62 condensates seems to be dependent on its oligomerization state and PTMs [93]. Thus, this mechanism, mediated by TRIM23/TBK1, could be investigated in the future to understand the molecular mechanism of ubiquitination of SQSTM1/p62 under HSV-1 infection in H4 cells, here reported.

Our analyses indicate that OPTN1 and NDP52 are also modulated by HSV-1 infection. As well asp62/SQSTM1 and NBR1, the function of OPTN1 can be modulated by phosphorylation and ubiquitination [94]. The molecular aspects behind the regulation of these selective autophagy receptors under HSV-1 infection should be studied, possibly by PTMs.

Furthermore, a remarkable finding in this investigation was the attenuated replication of HSV-1 in cells infected with the mutant strains R3616 and R3631. Unexpectedly, this decrease in growth, compared to strain F, is not due to an inhibition of viral protein synthesis; rather, we detected higher levels of viral intracellular proteins, compared to cells infected with HSV-1 strain F, which points to the crucial roles that ICP34.5 and US11 have to play in the HSV-1 pathogenesis. It has been reported that efficient viral replication depends on the function of ICP34.5, which is required to prevent the PKR response and consequently the translational shut-off [95]. Paradoxically, in our cell model (H4), the lack of ICP34.5 did not lead to translation arrest, and even so, resulted in a reduction in the growth of progeny by 3 orders of magnitude. This change in the replication efficiency was consistent with another previous investigations performed in mouse 3T6, SK-N-SH, CV-1, and MEF cells infected with HSV-1 strain R3616 or strain 1716 (strain 17+ ICP34.5 deletion mutant), with certain differences in the extent of the attenuation of virus growth between cell lines [95,96,97]. Whereas, for instance, the growth kinetics in BHK cells was indistinguishable from those infected with the parental strain [96], indicative that phenotype could be cell type dependent. In a similar way, the role of US11 in translational arrest seems to be dependent on cell type [98].We speculate that the H4 cell line may lack a cellular factor that is critical for egress of mutant HSV-1 particles, or alternatively, one hypothesis to reconcile these findings is that the ICP34.5 and US11 proteins may have additional functions required for egress of particles. This notion is supported by evidence obtained through electron microscopy, whereas the inefficient release of HSV-1 strain 1716 from infected 3T6 cells was demonstrated along with virus particle accumulation within nuclei and an important proportion of immature particles. This was presumably due to defaults in egress from the nuclei of the infected cell [96], phenotype in accordance with our finding of high intracellular levels of HSV-1 proteins in infected cells with the strain R3616. 

The importance of US11 in HSV-1 growth has been far less studied. This extent was explored using an in vitro model for Keratoconjunctivitis, human corneal epithelial cells (HCLEs). HCLEs cells infected with HSV-1 strain 17Δ11 (null mutant for US11) showed no translational arrest and demonstrated that US11 is dispensable for growth in vitro [98], in accordance with initial investigations, whereas the HSV-1 US11 gene was described as one non-essential gene [52,99]. US11 is dispensable for infection of several cell types derived from rodents and non-human primates [52], and it may play a role in the replication of HSV-1 in human cells subjected to heat shock [100]. However, in our investigation, it was notable that the lack of US11 generated a crosswise increase in intracellular HSV-1 proteins along with an attenuated growth. These dissenting results could open new perspectives about investigations performed with mutant strains since the conclusions must be carefully confined to the host cell that is characterized.

Finally, this research highlights that the autophagy process is a cellular mechanism that plays a dual role during the replication of HSV-1, constituting a key defense pathway for the host against the virus, which can be exploited by the virus to assure its replication.

## 5. Conclusions

HSV-1 is a highly prevalent human pathogen causing widespread clinical manifestations, which can be fatal. HSV-1 is a widely characterized virus, which has developed multiple strategies to evade the host defense mechanisms, leading to lifelong persistent infections. 

Collectively, the results of this research contribute to understanding the virus–host interaction by revealing that activating xenophagy in host cells is an intrinsic defense against HSV-1 (Model in Figure 10) capable of reducing the intracellular viral protein levels and progeny yield. Additionally, the autophagic receptors SQSTM1/p62, OPTN1, NBR1, and NDP52 could be degraded through xenophagy in the early stages of productive HSV-1 infection. The ability of HSV-1 to modulate the intrinsic host defense mechanisms, such as autophagy, emerges as critical strategies for its virulence. In this study, it is shown that HSV-1 could manipulate autophagy through an unsuspected role of the viral protein US11. It was found that SQSTM1/p62, OPTN1, NBR1, and NDP52, key proteins for the recognition of cargoes, are novel, reliable host factors targeted for HSV-1 protein US11. Moreover, the removal of the US11 protein from HSV-1 increases the ubiquitination of SQSTM1/p62, revealing a novel mechanism by which US11 could counteract antiviral functions, which has not been reported so far. Taken together, these data expand our understanding of the virus–host interaction and how HSV-1 evades host barriers to persist throughout the host’s lifetime, facilitating its spread over the human population.

This research, together with some previous studies [40,80], has begun to unveil that manipulation of autophagy may serve as a potential therapeutic strategy to counteract HSV-1 infection, where boosting the host defense mechanism of autophagy may be a promising way to fight against HSV-1 infection. 

## Figures and Tables

**Figure 1 cells-13-01256-f001:**
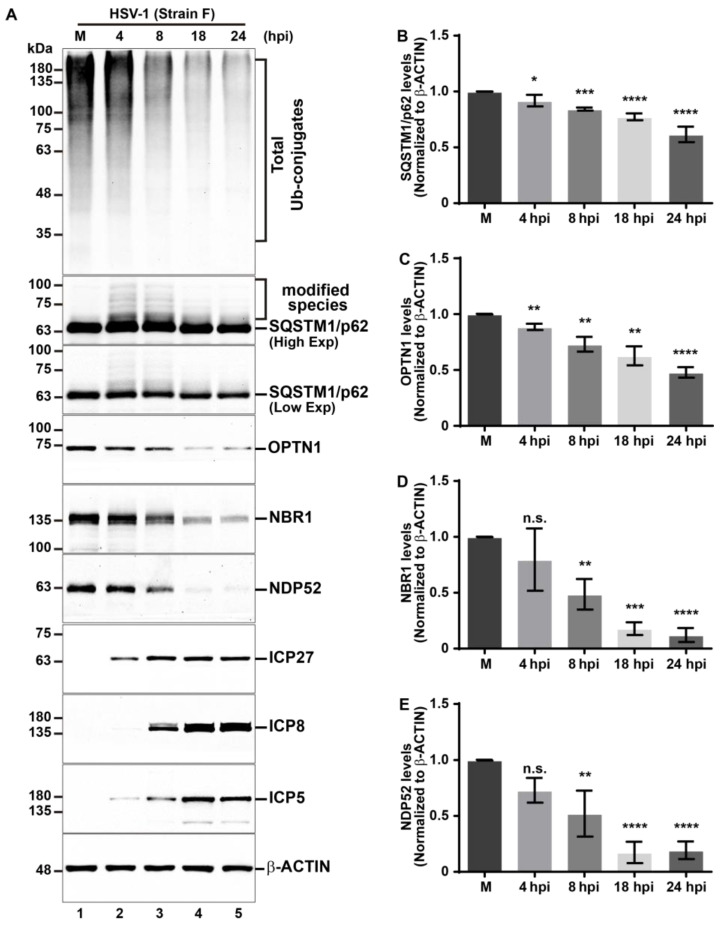
HSV-1 infection reduces the total levels of Ub conjugates and various autophagic receptors in H4 cells. H4 cells uninfected (M; mock) and infected with HSV-1 strain F at MOI 10 were analyzed at 4, 8, 18, and 24 h post-infection (hpi). (**A**) Protein extracts were subjected to immunoblot using monoclonal antibodies against total Ub conjugates, SQSTM1/p62, OPTN1, NBR1 and NDP52. Representative proteins of the replicative cycle of HSV-1 were detected with monoclonal antibodies against ICP27 (immediate early protein), ICP8 (early protein), and ICP5 (late protein) to control infection. β-ACTIN was used as a loading control. Positions of molecular weight markers (kDa) are indicated on the left. A representative image from immunoblot was displayed in high exposure (High Exp) to indicate modified SQSTM1/p62 species, in comparison to the image captured at low exposure (Low Exp). Densitometric quantification of (**B**) SQSTM1/p62, (**C**) OPTN1, (**D**) NBR1, and (**E**) NDP52 protein levels were normalized to β-ACTIN. Statistical significance was determined by One-Way ANOVA followed by Tukey’s Test. The bars represent the mean ± SD of biological replicates (SQSTM1/p62 *n* = 4; OPTN1 *n* = 3; NBR1 *n* = 3; NDP52 *n* = 3); n.s., not significant, * *p* < 0.05, ** *p* < 0.01, *** *p* < 0.001, **** *p* < 0.0001.

**Figure 2 cells-13-01256-f002:**
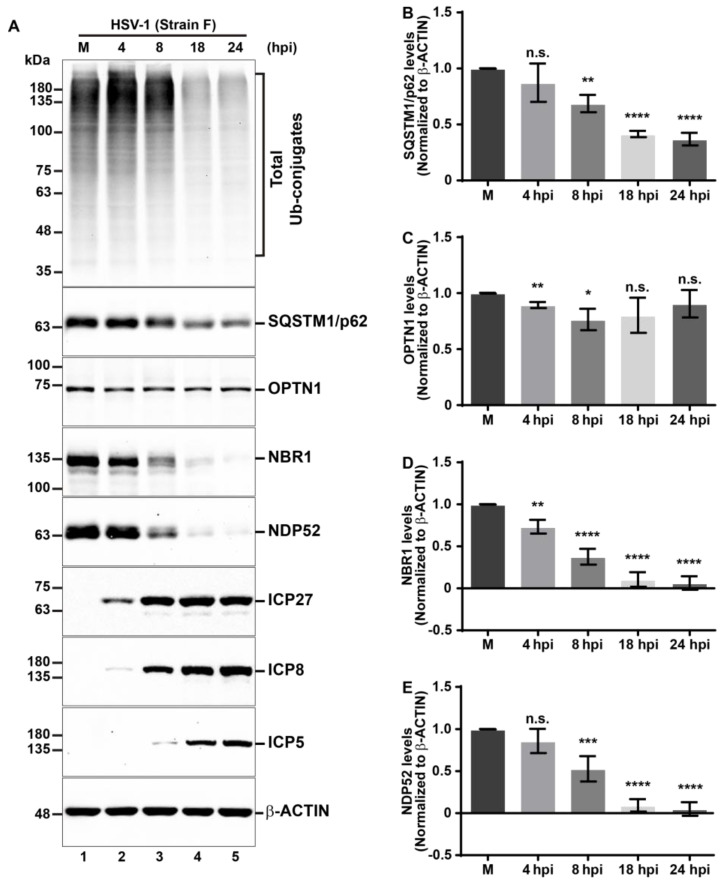
HSV-1 infection reduces SQSTM1/p62, OPTN1, NBR1 and NDP52 autophagic receptors levels in HaCaT cells. HaCaT cells uninfected (M; mock) and infected with HSV-1 strain F at MOI 10 were analyzed at 4, 8, 18, and 24 h post-infection (hpi). (**A**) Protein extracts were subjected to immunoblot using monoclonal antibodies against total Ub conjugates, SQSTM1/p62, OPTN1, NBR1, and NDP52. Representative proteins of the replicative cycle of HSV-1 were detected with monoclonal antibodies, ICP27 (immediate early protein), ICP8 (early protein), and ICP5 (late protein) to control infection. β-ACTIN was used as a loading control. Positions of molecular weight markers (kDa) are indicated on the left. Densitometric quantification of (**B**) SQSTM1/p62, (**C**) OPTN1, (**D**) NBR1, and (**E**) NDP52 protein levels were normalized to β-ACTIN. Statistical significance was determined by One-Way ANOVA followed by Tukey’s Test. The bars represent the mean ± SD of biological replicates (SQSTM1/p62 *n* = 4; OPTN1 *n* = 3; NBR1 *n* = 4; NDP52 *n* = 4); n.s., not significant; * *p* < 0.05, ** *p* < 0.01, *** *p* < 0.001, **** *p* < 0.0001.

**Figure 3 cells-13-01256-f003:**
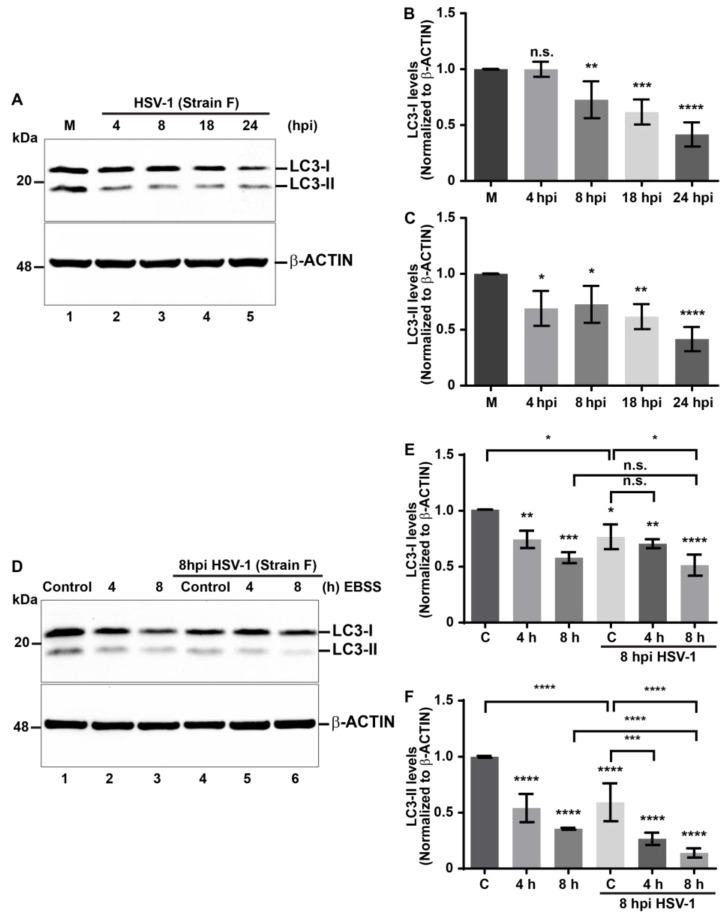
HSV-1-infected cells respond to EBSS by reducing the levels of the autophagic LC3-II marker. (**A**) H4 uninfected (M; mock) and infected with HSV-1 (strain F) at MOI 10 were analyzed at 4, 8, 18, and 24 (hpi). (**D**) H4 uninfected (M; mock) and infected with HSV-1 strain F at MOI 10 were cultured in medium-full supplemented for 8 h (control) or in Earle’s balanced salts solution (EBSS) for 4 or 8 h to induce autophagy. Protein extracts were subjected to immunoblot. (**A**,**D**) LC3 levels were assayed with a polyclonal antibody against LC3 that recognizes LC3-I and LC3-II. β-ACTIN was used as a loading control. Positions of molecular weight markers (kDa) are indicated on the left. Densitometric quantification of (**B**,**E**) LC3-I and (**C**,**F**) LC3-II was normalized to β-ACTIN. Statistical significance was determined by One-Way ANOVA followed by Tukey’s test. The bars represent the mean ± SD of biological replicates (LC3-I *n* = 4; LC3-II *n* = 4); n.s., not significant; * *p* < 0.05, ** *p* < 0.01, *** *p* < 0.001, **** *p* < 0.0001.

**Figure 4 cells-13-01256-f004:**
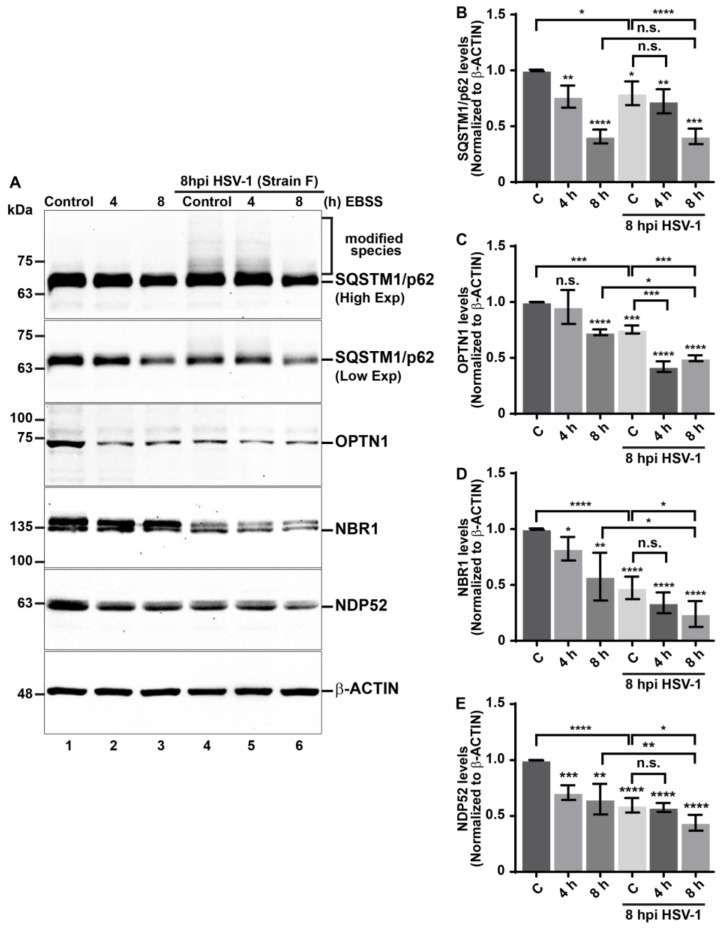
HSV-1-infected cells respond to EBSS by reducing the levels of OPTN1, NBR1, and NDP52 but not SQSTM1/p62. (**A**) H4 cells uninfected and infected with HSV-1 strain F at MOI 10 were cultured in medium-full supplemented for 8 h (control) or in Earle’s balanced salts solution (EBSS) for 4 or 8 h to induce autophagy. Protein extracts were subjected to immunoblot using monoclonal antibodies against SQSTM1/p62, OPTN1, NBR1, and NDP52. β-ACTIN was used as a loading control. Positions of molecular weight markers (kDa) are indicated on the left. A representative image from immunoblot was displayed in high exposure (High Exp) to indicate modified SQSTM1/p62 species, in comparison to the image captured at low exposure (Low Exp). Densitometric quantification of (**B**) SQSTM1/p62, (**C**) OPTN1, (**D**) NBR1, and (**E**) NDP52 protein levels were normalized to β-ACTIN. Statistical significance was determined by One-Way ANOVA followed by Tukey’s test. The bars represent the mean ± SD of biological replicates (SQSTM1/p62 *n* = 4; OPTN1 *n* = 3; NBR1 *n* = 4; NDP52 *n* = 4); n.s., not significant; * *p* < 0.05, ** *p* < 0.01, *** *p* < 0.001, **** *p* < 0.0001.

**Figure 5 cells-13-01256-f005:**
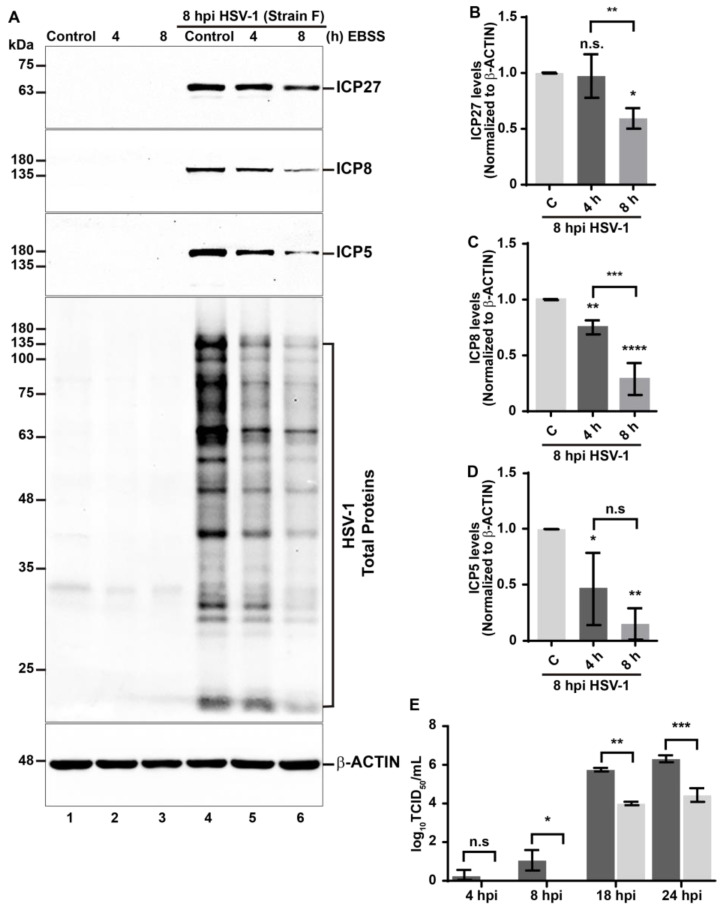
Starvation-induced autophagy reduces the intracellular levels of HSV-1 proteins and progeny yield. (**A**) H4 cells uninfected and infected with HSV-1 strain F at MOI 10, were cultured in full medium supplemented (control) for 8 h or in Earle’s balanced salts solution (EBSS) for 4 or 8 h to induce autophagy. Protein extracts were subjected to immunoblot using monoclonal antibodies against representative proteins of the replicative cycle of HSV-1, ICP27 (immediate early protein), ICP8 (early protein), and ICP5 (late protein), and a polyclonal antibody against HSV-1. β-ACTIN was used as a loading control. Positions of molecular weight markers (kDa) are indicated on the left. Densitometric quantification of (**B**) ICP27, (**C**) ICP8, and (**D**) ICP5 protein levels were normalized to β-ACTIN. Statistical significance was determined by One-Way ANOVA followed by Tukey’s test. The bars represent the mean ± SD of biological replicates (ICP27 *n* = 4; ICP8 *n* = 4; ICP5 *n* = 3); n.s., not significant; * *p* < 0.05, ** *p* < 0.01, *** *p* < 0.001, **** *p* < 0.0001. (**E**) Progeny HSV-1 yield was assessed by titration in Vero cells through TCID_50_ at 96 hpi. The graphic shows the viral titer of HSV-1 in the supernatants from H4 HSV-1-infected cells during 4, 8, 18, and 24 hpi in culture medium full supplemented (dark grey bars) or EBSS (light grey bars). Statistical significance was determined by Two-Way ANOVA followed by Sidak’s multiple comparisons test. The bars represent the mean ± SD of biological replicates (culture medium *n* = 3; EBSS *n* = 3); n.s., not significant; * *p* < 0.05, ** *p* < 0.01, *** *p* < 0.001.

**Figure 6 cells-13-01256-f006:**
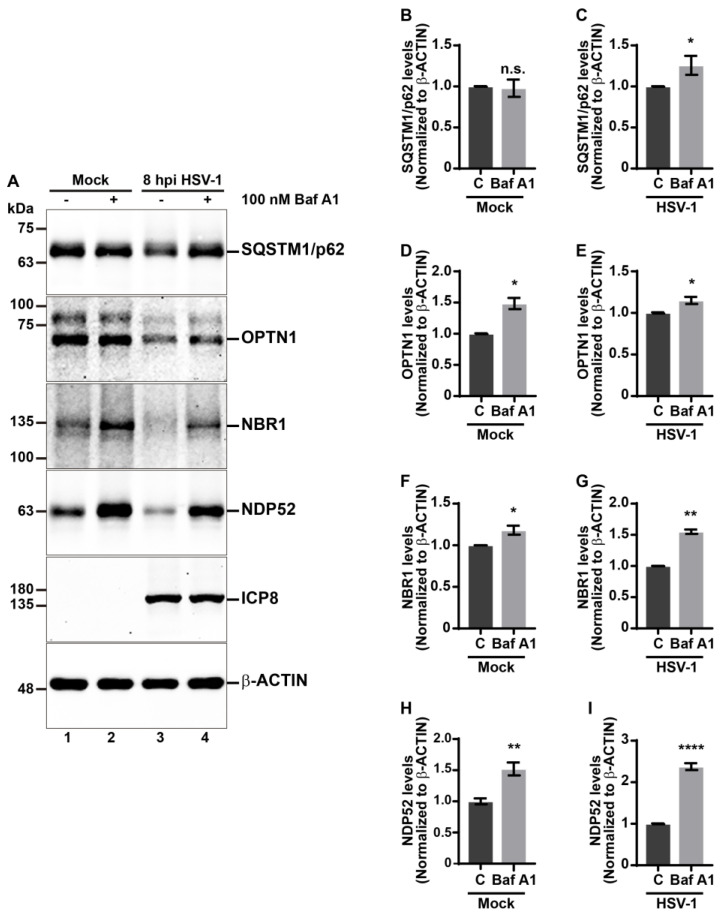
The degradation of autophagic receptors SQSTM1/p62, OPTN1, NBR1, and NDP52 during HSV-1 infection depends on the autophagic flux. H4 cells were either uninfected (mock) or infected with HSV-1 strain F at MOI 10. They were then treated with either vehicle DMSO (control) or 100 nM Baf A1 for 8 h to inhibit autophagic flux. Protein extracts were subjected to immunoblot using monoclonal antibodies against (**A**) SQSTM1/p62, OPTN1, NBR1, and NDP52. HSV-1 ICP8, an early protein, was used to control the infection. β-ACTIN was used as a loading control. Positions of molecular weight markers (kDa) are indicated on the left. Densitometric quantification of (**B**,**C**) SQSTM1/p62, (**D**,**E**) OPTN1, (**F**,**G**) NBR1, and (**H**,**I**) NDP52 protein levels were normalized to β-ACTIN. Statistical significance was determined by Student’s *t*-test. Bars represent means ± SD of biological replicates (SQSTM1/p62 *n* = 3; OPTN1 *n* = 3; NBR1 *n* = 3; NDP52 *n* = 3); n.s., not significant; * *p* < 0.05, ** *p* < 0.01, **** *p* < 0.0001.

**Figure 7 cells-13-01256-f007:**
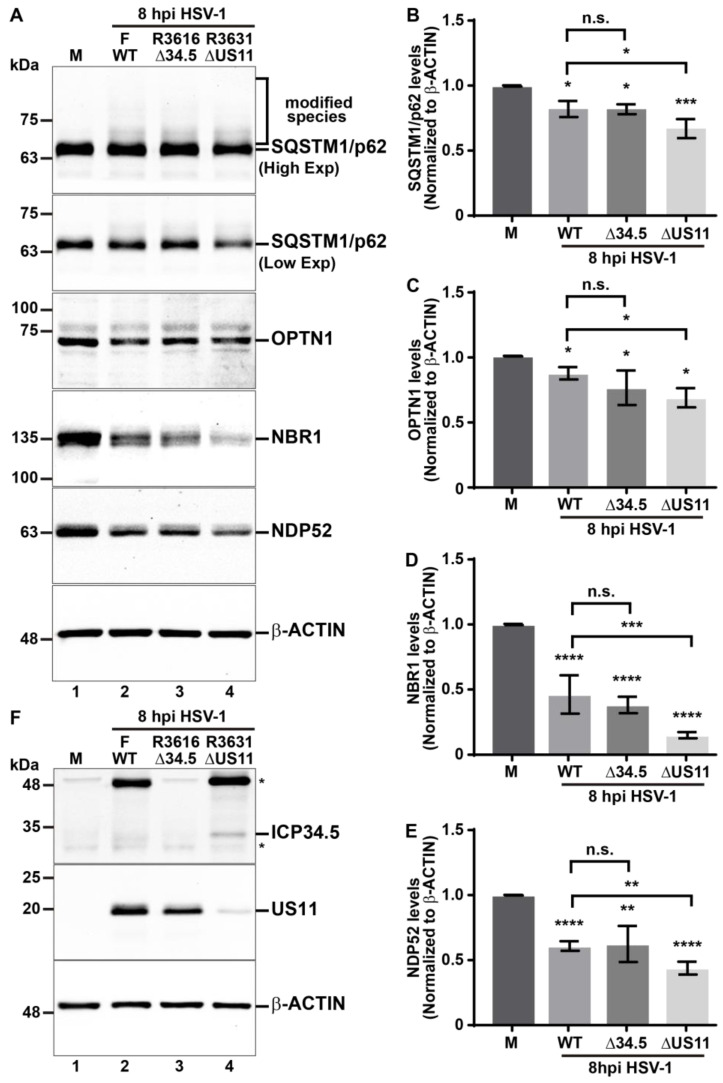
The tegument HSV-1 protein US11 restricts the degradation of autophagic receptors during HSV-1 infection. (**A**,**F**) H4 cells uninfected (M; mock) and infected with HSV-1 strain F (WT), HSV-1 strain R3616 null mutant for ICP34.5 (Δ34.5), and HSV-1 strain R3631 null mutant for US11 (ΔUS11) at MOI 10 for 8 h. Protein extracts were subjected to immunoblot using monoclonal antibodies against (**A**) SQSTM1/p62, OPTN1, NBR1, and NDP52. (**F**) The expression of viral proteins ICP34.5 and US11 was controlled using monoclonal antibodies. The asterisks appear in (**F**) to show unspecific bands recognized by the antibody. β-ACTIN was used in (**A**,**F**) as a loading control. Positions of molecular weight markers (kDa) are indicated on the left. A representative image from immunoblot in (**A**) was displayed in high exposure (High Exp) to indicate modified SQSTM1/p62 species, in comparison to the image captured at low exposure (Low Exp). Densitometric quantification of (**B**) SQSTM1/p62, (**C**) OPTN1, (**D**) NBR1, and (**E**) NDP52 protein levels was normalized to β-ACTIN. Statistical significance was determined by One-Way ANOVA followed by Tukey’s Test. Bars represent means ± SD of biological replicates (SQSTM1/p62 *n* = 3; OPTN1 *n* = 3; NBR1 *n* = 4; NDP52 *n* = 3); n.s., not significant; * *p* < 0.05, ** *p* < 0.01, *** *p* < 0.001, **** *p* < 0.0001.

**Figure 8 cells-13-01256-f008:**
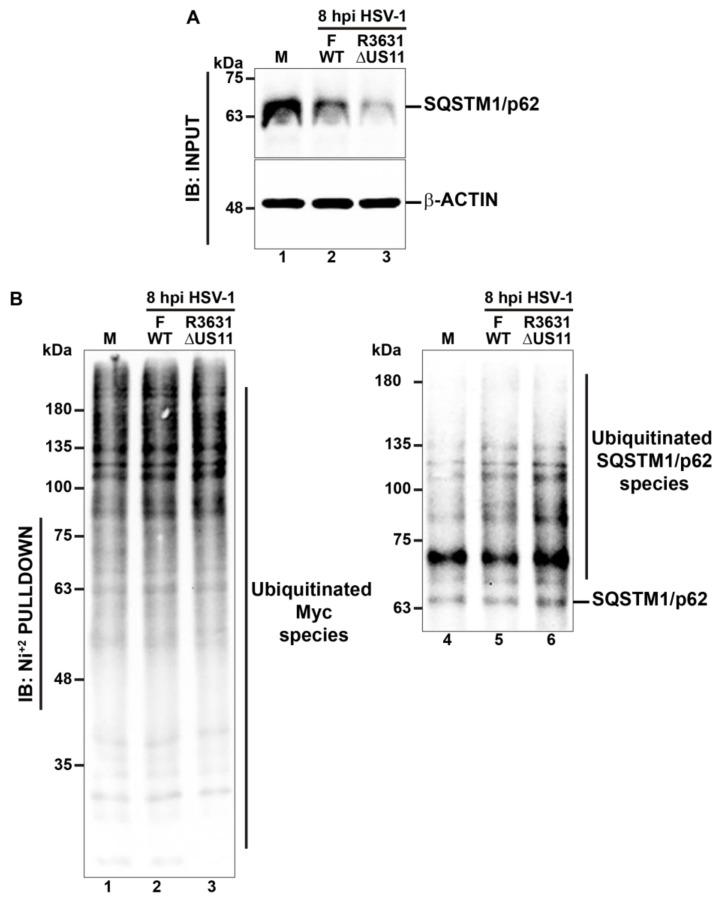
HSV-1 infection enhances the ubiquitination of SQSTM1/p62, a process that is counteracted by the tegument HSV-1 protein US11. His-Ub pulldown was assessed for the specific detection of Ub conjugates in SQSTM1/p62. H4 cells were transfected for 12 h with 6His-Myc-Ub, followed by infection with HSV-1 strain F (WT) and HSV-1 strain R3631 null mutant for US11 (ΔUS11) at MOI 10 for 8 hpi. (**A**) Unpurified samples derived from 10% of the cells used for the His-pulldown (IB: INPUT) were analyzed by immunoblot with a monoclonal antibody for the protein SQSTM1/p62. β-ACTIN was used as a loading control. (**B**) The total intracellular Ub conjugates were purified under denaturing conditions by affinity using a Ni^+2^ loaded agarose resin, and the purified fractions were analyzed by immunoblot (IB: Ni^+2^ PULLDOWN) with a monoclonal antibody against Myc-tag (clone 9B11), which was depicted as Ubiquitinated Myc species. Ub conjugated SQSTM1/p62 was detected with a monoclonal antibody against SQSTM1/p62 and was depicted as Ubiquitinated SQSTM1/p62 species. Positions of molecular weight markers (kDa) are indicated on the left.

**Figure 9 cells-13-01256-f009:**
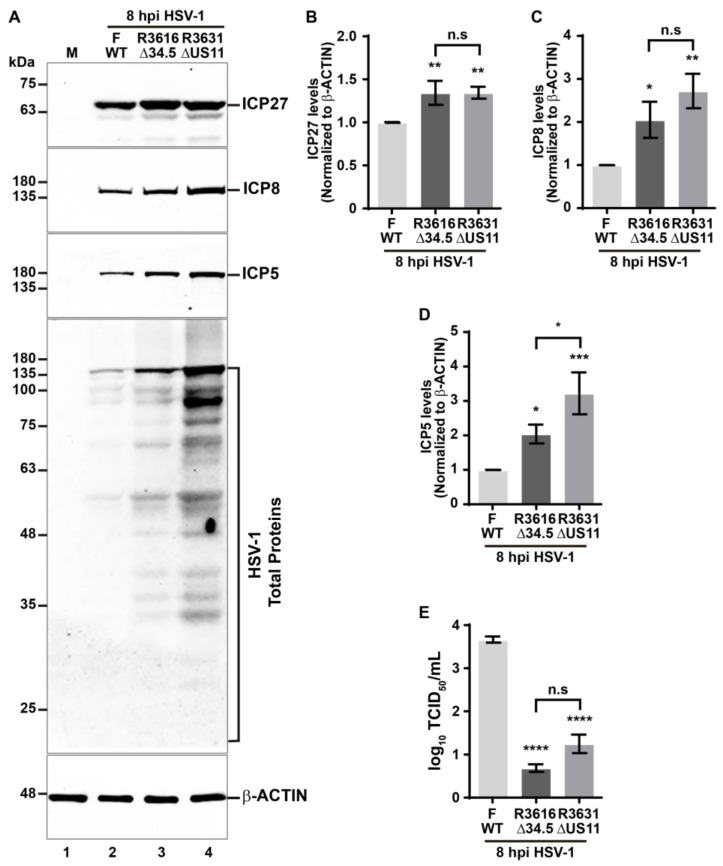
The removal of tegument HSV-1 protein US11 negatively impacts the progeny HSV-1 yield. (**A**) H4 cells uninfected (M; mock) and infected with HSV-1 strain F (WT), HSV-1 strain R3616 null mutant for ICP34.5 (Δ34.5), and HSV-1 strain R3631 null mutant for US11 (ΔUS11) at MOI 10 for 8 h. Protein extracts were subjected to immunoblot using monoclonal antibodies against representative proteins of the replicative cycle of HSV-1, ICP27 (immediate early protein), ICP8 (early protein), and ICP5 (late protein), and a polyclonal antibody against HSV-1. β-ACTIN was used as a loading control. Positions of molecular weight markers (kDa) are indicated on the left. Densitometric quantification of (**B**) ICP27, (**C**) ICP8, and (**D**) ICP5 protein levels was normalized to β-ACTIN. Statistical significance was determined by One-Way ANOVA followed by Tukey’s test. The bars represent the mean ± SD of biological replicates (ICP27 *n* = 3; ICP8 *n* = 3; ICP5 *n* = 3); n.s., not significant; * *p* < 0.05, ** *p* < 0.01, *** *p* < 0.001. (**E**) Progeny HSV-1 yield was assessed by titration in Vero cells through TCID_50_ at 96 hpi. The graphic shows the viral titer of HSV-1 in the supernatants from H4 cells infected with HSV-1 strain F (WT), HSV-1 strain R3616 null mutant for ICP34.5 (Δ34.5), and HSV-1 strain R3631 null mutant for US11 (ΔUS11). Statistical significance was determined by One-Way ANOVA followed by Tukey’s test. The bars represent the mean ± SD of biological replicates (*n* = 3); n.s., not significant; **** *p* < 0.0001.

**Figure 10 cells-13-01256-f010:**
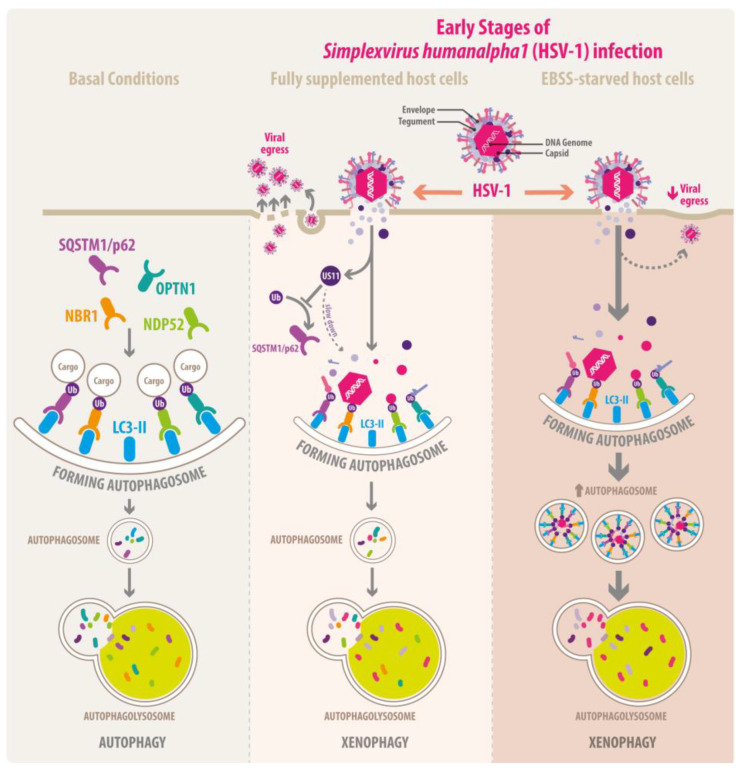
Schematic model of xenophagy in host cells as intrinsic defense against HSV-1 with the underlying clearance of the autophagic receptors. The model depicts the autophagy process in basal conditions (**left panel**) and in early stages of HSV-1 infection in fully supplemented (**middle panel**) and EBSS-starved host cells (**right panel**). During basal autophagy (**left panel**), the autophagic receptors such as SQSTM1/p62, OPTN1, NBR1, and NDP52 can interact with ubiquitinated cargo and mediate its recruitment into forming autophagosomes through LC3 binding. Once cargo has been engulfed, the external membrane of the autophagosome fuses with a lysosome for subsequent breakdown of the components in the autophagolysosome. In early stages of HSV-1 infection, detection of the virus by the host cell promotes the stimulation of autophagy. This pathway, referred to as Xenophagy (**middle panel**), may act as an intrinsic host defense against HSV-1, mediated by autophagic receptors to limit virus replication thereby reducing progeny egress of HSV-1. We propose that the tegument HSV-1 protein US11 slows down the autophagy, restricting the degradation of SQSTM1/p62, OPTN1, NBR1, and NDP52, a novel mechanism by which US11 could counteract autophagy, operating as a brake on this cellular process. Moreover, as part of the mechanism of viral modulation, during HSV-1 infection, we found that the ubiquitination of SQSTM1/p62 increases, but this PTM could be negatively modulated by US11 to ensure its replication. When the host cells are subjected to starvation (**right panel**), the defensive mechanism is improved, reducing the progeny yield of HSV-1 even further.

## Data Availability

The original contributions presented in the study are included in the article/Appendix A, further inquiries can be directed to the corresponding author.

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
