# Peer review of "An Intrinsic Host Defense against HSV-1 Relies on the Activation of Xenophagy with the Active Clearance of Autophagic Receptors"

_cells, 2024, doi:10.3390/cells13151256_

Round 1

Reviewer 1 Report (Previous Reviewer 1)

Comments and Suggestions for Authors

I have reviewed this revised manuscript and I believe the authors have addressed most of my concerns.

I now recommend this manuscript for publication in Cells.

Reviewer 2 Report (Previous Reviewer 2)

Comments and Suggestions for Authors

The manuscript has been improved and no further revisions are needed. In my opinion, it now warrants publication in Cells

This manuscript is a resubmission of an earlier submission. The following is a list of the peer review reports and author responses from that submission.

Round 1

Reviewer 1 Report

Comments and Suggestions for Authors

In this manuscript, the authors claim that HSV-1 infection induces a reduction in autophagic receptor levels in vitro. Although the manuscript is potentially interesting, I have several points that should be addressed by the authors. The specific points are as follows.

Major points.

1.     The authors claim that the expression levels of autophagic receptors were decreased by the autophagic machinery after viral infection. However, according to the supplementary data, autophagy was not the only mechanism for the downregulation of these proteins, but the transcriptional and translational steps were strongly affected. In Figure 5, Baf A1 treatment did not sufficiently rescue the downregulation of these proteins. Therefore, I think this point should be revised and the text should be changed.

2.     The authors claim that autophagic receptor clearance is a key step in the host's intrinsic antiviral defense. Indeed, the US11 mutant promoted the downregulation of autophagic receptor proteins, but there is no evidence as to how much this mutant affects viral infectivity through clearance of the proteins. This should be addressed.

3.     The authors claim that HSV-1 promotes selective autophagy rather than bulk autophagy (Abstract). The authors should reasonably explain that "HSV-1 promotes selective autophagy". What data are they referring to?

Minor points.

1.     It is better to illustrate findings as a figure.

2.     Figure 2: Ub data should also be provided in HaCaT cells.

3.     English should be carefully edited by a professional English editing service.

Reviewer 2 Report

Comments and Suggestions for Authors

The authors investigate the role of US11 protein of HSV-1 on xenophagy by accelerating the degradation of the autophagic receptors SQSTM1/p62, OPTN1, NBR1, and NDP52 in early stages of productive HSV-1 infection.  This paper contributes to the current literature and brings new information to the field. In fact, the submitted work is original and gives important insights of a particular aspect of HSV-1 infection in neuronal and epithelial cells.

The manuscript is difficult to understand and  a careful revision of the text is suggested, in order to facilitate the reader's overall understanding of the various aspects presented in the manuscript, using shorter sentences and resuming each finding at the end of each paragraph. I would recommend the authors to process the paper through an English revision, to check language, readability, and grammar used in the manuscript.

The manuscript provided strong evidence and a significant discussion of the data, but it lacks a synthetic resume and conclusion.

Specific comments to the authors are as follow:

Acronym should be explained as it’s the first time it appears in the text.

All figures: I suggest choosing a unique name for the y-axis unit in the densitometric analysis graphs.  You could edit the graphs in Figures 5B, 5C, 5D, 5E, 5F,5G, 5H, 5I by adding "Arbitrary Units" instead of "A. U."

Figure 1- You might add viral controls (ICP5 to ICP27) on H4 cells (Figure 1), as you did for HaCat cells (Figure 2). 

Line 177 – Several concentrations… Which ones?

Lines 285 – 289 – the differences in the washing procedures are not clear. Try to summarize the paragraph (and use 2 or two).

Figure 3 A/C – LC3-II 8hpi quantification (in figure 3C) does not completely correlate to the density of the band (figure 3A). You might want to choose a more representative picture to replace the figure 3A.

Line 406- “y” instead “and”

Figures 5B, 5C, 5D, 5E, 5F,5G, 5H, 5I: You should keep the same scale of values.

Figures 8B, 8C, 8D: You should keep the same scale of values.

Line 803 - …opposite to some previous reports. The reference is missing.